# Double Descent in Adversarial Training: An Implicit Label Noise Perspective

## Abstract

Here, we show that the robust overfitting shall be viewed as the early part of an epoch-wise double descent — the robust test error will start to decrease again after training the model for a considerable number of epochs. Inspired by our observations, we further advance the analyses of double descent to understand robust overfitting better. In standard training, double descent has been shown to be a result of label flipping noise. However, this reasoning is not applicable in our setting, since adversarial perturbations are believed not to change the label. Going beyond label flipping noise, we propose to measure the mismatch between the assigned and (unknown) true label distributions, denoted as *implicit label noise*. We show that the traditional labeling of adversarial examples inherited from their clean counterparts will lead to implicit label noise. Towards better labeling, we show that predicted distribution from a classifier, after scaling and interpolation, can provably reduce the implicit label noise under mild assumptions. In light of our analyses, we tailored the training objective accordingly to effectively mitigate the double descent and verified its effectiveness on three benchmark datasets.

## 1 Introduction

In adversarial training, a typical phenomenon is that after a certain training epoch, the robust test error will start to increase constantly with further training, despite the robust training error continues to decrease (Rice et al., 2020). This phenomenon, known as *robust overfitting*, is believed to be separated from *double descent* (Belkin et al., 2019) in the literature (Rice et al., 2020).

Here, we find that robust overfitting shall be viewed as the early part of an epoch-wise double descent (Nakkiran et al., 2020). As shown in Figure 1, for relatively large models, the robust test error increases only transiently and will eventually decrease again after a considerable number of training epochs. Moreover, we observe consistent and similar patterns across various training settings (e.g., different sample sizes, optimizers, learning rate schedulers, and neural architectures), which further verifies our intuition on the connection between the robust overfitting and the double descent (See Appendix B.1).

Inspired by this observation, we further analyze the epoch-wise double descent to better understand the robust overfitting. In standard training, the double descent is shown to be related to the label flipping noise (Nakkiran et al., 2020; Yang et al., 2020b). However, adversarial perturbations are believed to not change underlying labels, which is further con-

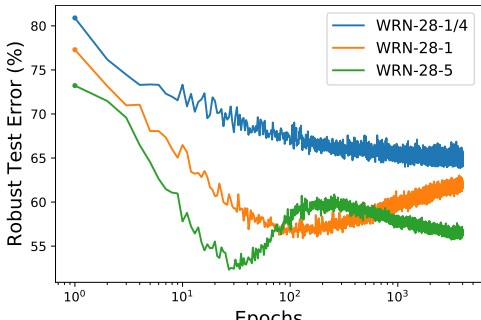

Figure 1: Robust overfitting can be viewed as an early part of the epoch-wise double descent. Here we employ PGD training (Madry et al., 2018) on CIFAR-10 (Krizhevsky, 2009) with Wide ResNet (WRN) (Zagoruyko & Komodakis, 2016) and a fixed learning rate.

firmed empirically from our manual inspection. Therefore, the popular label flipping noise explanation is not applicable here. On the other hand, it is reasonable to believe that adversarial perturbation distorts the label distribution since the examples do become more ambiguous after perturbation. Thus, we focus on the *implicit label noise* which measures the mismatch between the assigned label

distribution and (unknown) true label distribution. Extensive analyses of such implicit label noise reveal the intriguing dependency of robust overfitting on perturbation size (Dong et al., 2021b) and data quality (Dong et al., 2021a).

Guided by our analyses, we design a theoretically-grounded method to mitigate the robust overfitting. The key idea is to resort to an alternative labeling of the adversarial examples. We show that the predictive label distribution of a probabilistic classifier adversarially trained as usual, but after being properly scaled and interpolated, can be utilized as a better labeling of the adversarial examples and provably reduce the implicit label noise. This echoes the recent empirical practice of incorporating knowledge distillation (Hinton et al., 2015) into adversarial training (Chen et al., 2021). While previous works intuitively select fixed scaling and interpolation parameters for knowledge distillation, we show that it is possible to fully unleash the potential of knowledge distillation by automatically determining the set of parameters that maximally reduces the implicit label noise with a strategy similar to confidence calibration (Guo et al., 2017). Such strategy can further mitigate robust overfitting to a minimal amount without additional human tuning effort. Extensive experiments on different datasets, training methods, neural architectures and robustness evaluation metrics verify the effectiveness of our method.

In summary, our findings and contributions are as follows.

- We show that robust overfitting shall be viewed as the early part of an epoch-wise double descent, extending the common belief in adversarial training.
- We show that double descent in adversarial training may originate from the implicit label noise introduced by improper labeling of adversarial examples in adversarial training practice.
- We show an alternative labeling of the adversarial examples can be established to provably reduce the implicit label noise and mitigate the robust overfitting.

The remainder of this paper is organized as follows. In Section 2, we briefly review the existing works that explore robust overfitting and double descent. In Section 3, we theoretically explore the origin of double descent in adversarial training from an implicit label noise perspective. In Section 4, we propose to mitigate the double descent in adversarial training by alternative labeling based on our understanding. Section 5 demonstrates the effectiveness of our method on realistic datasets. Conclusions and further implications are discussed in Section 6.

## 2 RELATED WORK

**Robust overfitting and double descent in adversarial training.** Double descent refers to the phenomenon that overfitting by increasing model complexity will eventually improve test set performance (Neyshabur et al., 2017; Belkin et al., 2019). This appears to conflict with the robust overfitting phenomenon in adversarial training, where increasing model complexity by training longer will impair test set performance constantly after a certain point during training. It is thus believed in the literature that double descent and robust overfitting are separate phenomena (Rice et al., 2020).

Towards a more complete understanding of robust overfitting, in this work, we conduct adversarial training for exponentially more epochs than the typical practice. We find that robust overfitting shall be viewed as the early part of an epoch-wise double descent. And increasing the model architecture size, another way to increase the model complexity, can modulate the epoch-wise double descent curve such that either the overfitting curve is shown, or the entire double descent curve is revealed within the same number of training epochs as shown in Figure 1. Therefore, robust overfitting will not go beyond modern generalization theory as an exception and should be adequately explained by the origin of double descent such as label noise.

A recent work also considers a different notion of double descent that is defined with respect to the perturbation size (Yu et al., 2021). Such double descent might be more related to the robustness-accuracy trade-off problem (Papernot et al., 2016; Su et al., 2018; Tsipras et al., 2019; Zhang et al., 2019), rather than the classic picture of double descent based on model complexity.

**Understand double descent in adversarial training.** In standard training, double descent is often attributed to increased variance, with label noise being a common source. Definitions of label noise vary in literature. Theoretically-grounded analyses of double descent focus on additive label noise (Advani & Saxe, 2020; Mei & Montanari, 2019; Hastie et al., 2019; Belkin et al., 2020; d'Ascoli

et al., 2020), but only applicable to regression problems. Theoretical results on double descent are scarce on classification problems, with a few works introducing noise by randomly masking the feature vector (Deng et al., 2019; Kini & Thrampoulidis, 2020). Analyses of double descent on classification problems are more common in empirical studies, where a typical way to induce double descent is to inject label flipping noise, namely the labels of a random fraction of training examples are flipped to other labels (Nakkiran et al., 2020; Yang et al., 2020b). However, such a definition of label noise cannot properly fit the scenario in adversarial training, where labels are not likely to be flipped due to small adversarial perturbation.

To understand the double descent in adversarial training within the existing picture of double descent, we introduce implicit label noise which is incurred by the mismatch between the assigned label distribution and true label distribution of the adversarial example. Such label noise can be interpreted as an instance-dependent, class dependent label noise in reference to the systematic studies on the taxonomy of label noise (Frénay & Verleysen, 2014; Song et al., 2020), thus can be the origin of double descent in adversarial training.

**Mitigate robust overfitting.** Robust overfitting hinders the practical deployment of adversarial training methods as the final performance is often sub-optimal. Various regularization methods including classic approaches such as $\ell_1$ and $\ell_2$ regularization and modern approaches such as cutout (Devries & Taylor, 2017) and mixup (Zhang et al., 2018) have been attempted to tackle robust overfitting, whereas they are shown to perform no better than simply early stopping the training on a validation set (Rice et al., 2020). However, early stopping raises additional concern as the best checkpoint of the robust test accuracy and that of the standard accuracy often do not coincide (Chen et al., 2020), thus inevitably sacrificing the performance on either criterion. Various regularization methods specifically designed for adversarial training are thus being proposed to outperform early stopping, including regularization the flatness of the weight loss landscape (Wu et al., 2020; Stutz et al., 2021), introducing low-curvature activation functions (Singla et al., 2021) and adopting stochastic weight averaging (Izmailov et al., 2018) and knowledge distillation (Hinton et al., 2015) (Chen et al., 2021). These methods are likely to suppress the implicit label noise, with the self-distillation framework (i.e. the teacher shares the same architecture as the student model) introduced by (Chen et al., 2021) as a particular example since introducing teacher's outputs as supervision is almost equivalent to the alternative labeling inspired by our understanding of the origin of label noise in adversarial training.

## 3 DOUBLE DESCENT FROM AN IMPLICIT LABEL NOISE PERSPECTIVE

In this section, we present a novel perspective to understand the double descent in adversarial training. All proofs in the remainder of this paper are provided in the appendix.

### 3.1 PRELIMINARIES

Let $\mathcal{X} \subset \mathbb{R}^d$ define the input space equipped with a norm $\|\cdot\| : \mathcal{X} \to \mathbb{R}^+$ and $\mathcal{Y} = \{1, \ldots, c\}$ define the label space. Considering a classification problem with a training set $\mathcal{D} = \{(x, y)\}$, where $x \in \mathcal{X}$ is an input and $y \in \mathcal{Y}$ is its assigned label. We denote $p(Y = j|x)$ as the distribution of the assigned labels of $x$ given by a group of annotators. Typically $y = \arg \max_j p(Y = j|x)$. When there is no ambiguity, we will denote $p(Y = j|x)$ as $p(y|x)$ with a slight abuse of notation. Let $y^* \in \mathcal{Y}$ be the true label of $x$ and $p(y^*|x)$ be its distribution, which can be given by a group of experts. Note $y^* = \arg \max_j p(Y^* = j|x)$.

**Assumption 3.1.** *We assume there is no label noise in $\mathcal{D}$, namely $y = y^*$ and $p(y|x) = p(y^*|x)$.*

**Definition 3.1** (Data quality). *Given an example $x \in \mathcal{X}$, we define its data quality as $\max_j p(Y^* = j|x)$, namely the maximum class probability of the true label distribution.*

**Assumption 3.2.** *We assume the original clean dataset $\mathcal{D}$ contains mostly high-quality data, namely $\max_j p(Y^* = j|x) \approx 1$.*

We now briefly review adversarial training. Let $f(\cdot; \theta) : \mathcal{X} \to \mathcal{Y}$ be a probabilistic classifier and $f(\cdot; \theta)_j$ be its $j$-th class probability. Adversarial training can be viewed as an data augmentation technique that trains the classifier $f$ on a set of adversarial examples $\mathcal{D}_\delta = \{(x_\delta, y_\delta)\}$, namely

$$\theta^* = \arg \min_\theta \mathbb{E}_{\mathcal{D}_\delta} \ell(f(x_\delta; \theta), y_\delta). \tag{1}$$

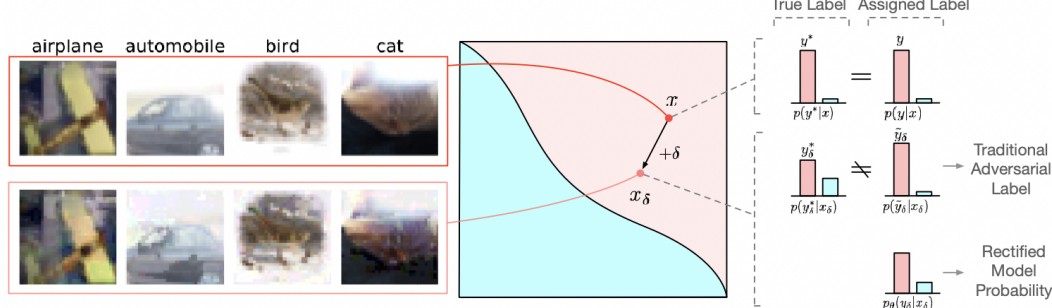

Figure 2: The illustration of the origin of implicit label noise in adversarial training. The traditional adversarial label introduces the implicit label noise by inducing a distribution mismatch between the assigned label distribution and true label distribution of the adversarial example. Our rectified model probability as an alternative labeling can provably reduce this distribution match.

Here, $x_\delta \equiv x + \delta$ is the adversarial example of $x$ produced by the inner maximization, where

$$\delta = \arg\max_{\delta, \|\delta\| \le \epsilon} \ell(f(x + \delta; \theta), y), \tag{2}$$

and $y_\delta \in \mathcal{Y}$ is the assigned label of $x_\delta$. Finally, we denote $y_\delta^* \in \mathcal{Y}$ as the true label of $x_\delta$.

**Remark 3.1.** *In adversarial training, it is the common practice that copies the label of a clean example to its adversarial counterpart, namely $y_\delta \leftarrow y$ and $p(y_\delta|x_\delta) \leftarrow p(y|x)$.*

We denote $\tilde{y}_\delta \equiv y$ and its associated distribution as the *traditional adversarial label*. We believe traditional adversarial label is the key problem that induces double descent in adversarial training.

### 3.2 TRADITIONAL ADVERSARIAL LABEL **DOES NOT INTRODUCE LABEL FLIPPING NOISE**

In standard learning, it is often necessary to manually inject label noise to make the double descent evident for modern neural architectures (Nakkiran et al., 2020; Yang et al., 2020b). We wish to check if the traditional adversarial label produces any label flipping noise, a typical type of label noise, namely if $\tilde{y}_\delta \ne y_\delta^*$ for any adversarial example $x_\delta$. This is equivalent to checking if $y^* \ne y_\delta^*$, since $\tilde{y}_\delta = y$ by the construction of the traditional adversarial label (Remark 3.1) and $y = y^*$ by the construction of our clean dataset (Assumption 3.1). Note that $y^* \ne y_\delta^*$ means the semantics of the adversarial example is distorted significantly such that its true label is now different from the true label of its clean counterpart.

We first visually check the adversarial examples created by the inner maximization in adversarial training. We are specifically interested in those examples with the lowest data quality as they are mostly likely to become label noise after perturbation. We estimate the data quality based on model ensemble as shown in Appendix E.1. One can find that the adversarial counterparts of those low-quality examples still match their original labels, albeit being slightly more ambiguous.

We have also found some evidence from the literature. Yang et al. (2020a) draw the same conclusion by studying the distance between examples in the input space. In the CIFAR-10 training set, they found that the minimum distance between any two examples from two different classes respectively is around $50/255$ in terms of $\ell_\infty$ norm, while the perturbation radius typically used in adversarial training is around $8/255$, which is significantly smaller. It is thus reasonable to claim that a significant amount of label flipping noise will not be produced by the inner maximization, and subsequently it cannot explain the double descent observed in adversarial training.

### 3.3 TRADITIONAL ADVERSARIAL LABEL **INTRODUCES IMPLICIT LABEL NOISE**

Although traditional adversarial label is unlikely to directly produce label flipping noise, we show that it can produce label noise implicitly.

An adversarial example often contains salient characteristics of classes other than the original label from human perspective (Tsipras et al., 2019; Ilyas et al., 2019). This implies that the true label distribution of an adversarial example is likely to be different from its clean counterpart, which means

$p(y_\delta^*|x_\delta) \neq p(y^*|x)$. Note this will not conflict with $y_\delta^* = y^*$ as discussed in Section 3.2, since the latter only implies $\arg\max_j p(Y_\delta^* = j|x_\delta) = \arg\max_j p(Y^* = j|x)$.

Now we show the traditional adversarial label is improper in terms of the label distribution. As illustrated in Figure 2, we have $p(\tilde{y}_\delta|x_\delta) = p(y|x)$ by Remark 3.1 and $p(y|x) = p(y^*|x)$ by Assumption 3.1, which together with the effect of adversarial perturbation $p(y_\delta^*|x_\delta) \neq p(y^*|x)$ imply that $p(\tilde{y}_\delta|x_\delta) \neq p(y_\delta^*|x_\delta)$. This means there is a distribution mismatch between the true label distribution and the assigned label distribution of the adversarial example. Such distribution mismatch introduces *implicit label noise* in the training set $\mathcal{D}_\delta$ employed in adversarial training.

**Definition 3.2** (Implicit label noise). *Given an example $x \in \mathcal{X}$, we define the implicit label noise as the probability that its assigned label [1] is different from its true label, namely $p(Y \neq Y^*|x)$.*

**Remark 3.2.** *Implicit label noise is equivalent to instance-dependent and class-dependent label noise, since $p(Y \neq Y^*|x) = 1 - \sum_j p(Y = j|Y^* = j, x)p(Y^* = j|x)$, and $p(Y \neq j|Y^* = j, x) = 1 - p(Y = j|Y^* = j, x)$ is a typical definition of label noise. It can easily seen that if $p(Y \neq Y^*|x) > 0$, $p(Y \neq j|Y^* = j, x) > 0$ for some $j$.*

Before showing the connection between the distribution mismatch and the *implicit label noise*, we need to quantify the distribution mismatch first. Here we adopt the *total variation distance*.

**Definition 3.3** (Total variation (TV) distance). *Let $J$ be a subset of the label sample space $\mathcal{Y}$. For two discrete probability distributions $p(y)$ and $p(y')$ where $y, y' \in \mathcal{Y}$, the total variation distance between them can be defined as*

$$\|p(y) - p(y')\|_{TV} = \max_J \left| \sum_{j \in J} p(y = j) - \sum_{j \in J} p(y' = j) \right| \tag{3}$$

We are now ready to present our main result.

**Theorem 3.1** (Traditional adversarial label incurs implicit label noise). *The implicit label noise incurred by the distribution mismatch between the traditional adversarial label and the true label of the adversarial example is lower-bounded, namely*

$$p(\tilde{y}_\delta \neq y_\delta^*|x_\delta) \geq \|p(\tilde{y}_\delta|x_\delta) - p(y_\delta^*|x_\delta)\|_{TV} = \|p(y^*|x) - p(y_\delta^*|x_\delta)\|_{TV}. \tag{4}$$

An informal proof can be sketched from a frequentist's view and help the understanding of implicit label noise. Say there are $M$ identical copies of $x_\delta$ in the training set $\mathcal{D}_\delta$, with their true labels and traditional adversarial labels distributing according to $p(y_\delta^*|x_\delta)$ and $p(\tilde{y}_\delta|x_\delta)$, respectively. The number of copies that have the same true label and assigned label is $M \sum_j \min\{p(\tilde{Y}_\delta = j|x_\delta), p(Y_\delta^* = j|x_\delta)\}$. The fraction of label noise exists in $\mathcal{D}_\delta$ is thus $1 - \sum_j \min\{p(\tilde{Y}_\delta = j|x_\delta), p(Y_\delta^* = j|x_\delta)\} = \|p(\tilde{y}_\delta|x_\delta) - p(y_\delta^*|x_\delta)\|_{TV}$ by the definition of the total variation distance.

## 3.4 DEPENDENCE OF IMPLICIT LABEL NOISE IN ADVERSARIAL TRAINING

We now show the implicit label noise in adversarial training depends on the perturbation size and the data quality, which will subsequently affect the double descent curves.

We consider optimal adversarial perturbation for simplicity, which is defined as

$$\delta^* = \arg\max_{\delta, \|\delta\| \leq \varepsilon} \ell(f^*(x + \delta; \theta), y), \tag{5}$$

where $f^*(\cdot)$ denotes the optimal probabilistic classifier such that $f^*(x) = p(y^*|x)$. With $\ell$ being the cross-entropy loss, the above equation can be rewritten as

$$\delta^* = \arg\min_{\delta, \|\delta\| \leq \varepsilon} f^*(x + \delta; \theta)_{y^*}, \tag{6}$$

which implies the optimal perturbation will reduce the probability mass of the true label. With the optimal adversarial perturbation, the minimum implicit label noise can be solved as follows.

---

[1]Note here the assigned label is a random variable rather than an outcome. See Theorem 3.1 for a better understanding of implicit label noise.

**Theorem 3.2** (Minimum implicit label noise under optimal perturbation). *Assume the true label distribution $p(y^*|x)$ is locally convex around $x$ and can be asymptotically described as*

$$\|\nabla_x \, p(Y^* = j|x)\| \propto 1 - p(Y^* = j|x), \quad p(Y^* = j|x) \to 1 \tag{7}$$

*we have*

$$\underline{\min} \, p(\tilde{y}_\delta \neq y_\delta^* | x_\delta) \propto \varepsilon(1 - \max_j p(Y^* = j|x)),$$

*where $\underline{\min}$ means the lower bound of the minimum label noise.*

In Appendix A.1, we show that Assumption (7) holds true for a Gaussian mixture model. The above theorem shows that, when the data quality of the clean example is relatively high, the implicit label noise introduced by adversarial perturbation is proportional to (1) the perturbation radius (2) the data quality. We conduct controlled experiments and empirically verify this correlation in Appendix B.2.

Interestingly, the above analysis implies that if the perturbation does not reduce the probability mass of the true label, it will not cause implicit label noise even with an extremely large perturbation radius. We demonstrate this in detail and empirically verified it for Gaussian noise in Appendix B.3.

### 3.5 IMPLICIT LABEL NOISE INDUCES DOUBLE DESCENT

In standard training, the effect of label noise on double descent has been rigorously studied based upon both analytical settings (Mei & Montanari, 2019; Hastie et al., 2019; Deng et al., 2019; Belkin et al., 2020) and bias-variance analyses (Jacot et al., 2020; Yang et al., 2020b; d'Ascoli et al., 2020). Since implicit label noise is just a special case of label noise (Remark 3.2), and adversarial training can be viewed as standard training on an augmented dataset (Equation (1)), it can be inferred that implicit label noise will increase the variance and make an evident double descent in adversarial training. To demonstrate this in a straightforward way, in Figure 3 we employ standard training on a dataset augmented by fixed adversarial perturbation and show it can indeed produce double descent.

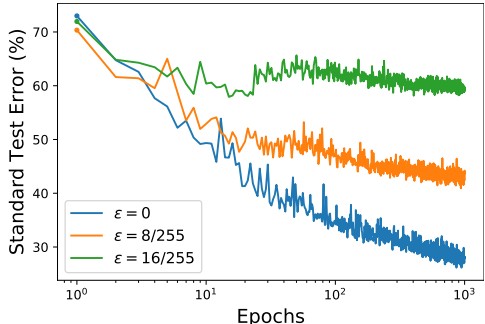

Figure 3: Without explicitly injecting label noise, standard training can also produce double descent if conducted on a dataset augmented by fixed and small adversarial perturbation, which can be properly explained by our implicit label noise perspective. Detailed experiment settings can be found in Appendix F.3.

Since implicit label noise modulates the double descent, and by Theorem 3.2 it depends on the perturbation radius and data quality, the double descent in adversarial training should strongly correlate with the perturbation radius and data quality. Indeed, it has been observed respectively that small perturbation radius will not induce robust overfitting (Dong et al., 2021b), and high-quality data will not induce robust overfitting (Dong et al., 2021a).

## 4 MITIGATE DOUBLE DESCENT IN ADVERSARIAL TRAINING

In this section, we focus on suppressing the implicit label noise in adversarial training to mitigate double descent from both theoretical and practical perspectives. Since the implicit label noise is incurred by the traditional adversarial label which improperly labels the adversarial example and causes a distribution mismatch between its assigned label and true label, we wish to find an alternative label (distribution) for the adversarial example to reduce such distribution mismatch.

### 4.1 RECTIFY MODEL PROBABILITY TO REDUCE DISTRIBUTION MISMATCH

We show that it is possible to reduce the distribution mismatch by utilizing the predictive probability of a classifier trained on traditional adversarial labels, which we will refer as the *model probability* for simplicity. We now provide a theoretical guarantee to show that, with *temperature scaling* (Hinton et al., 2015; Guo et al., 2017) enabled in the softmax function, model probability induces a distribution mismatch provably smaller than the traditional adversarial label.

**Theorem 4.1** (Model probability induces smaller distribution mismatch than the traditional adversarial label). *Let $f(x; \theta, T)$ denote the predictive probability of a hypothesis classifier scaled by temperature $T$, namely*

$$f(x; \theta, T)_j = \frac{\exp(z_j/T)}{\sum_j \exp(z_j/T)},$$

*where $z$ is the logits of the classifier from $x$.*

*Let $x_\delta$ be an adversarial example correctly classified by a classifier $f$, i.e. $\arg\max_j f(x_\delta; \theta)_j = y_\delta^*$, then there exists $T$, such that*

$$\|f(x_\delta; \theta, T) - p(y_\delta^*|x_\delta)\|_{TV} \leq \|p(\tilde{y}_\delta|x_\delta) - p(y_\delta^*|x_\delta)\|_{TV}.$$

One possible way to further reduce the distribution mismatch is to interpolate between the model probability and the traditional adversarial label.

**Theorem 4.2** (Interpolation can further reduce the distribution mismatch). *Let $x_\delta$ be an adversarial example incorrectly classified by a classifier $f$, i.e. $\arg\max_j f(x_\delta; \theta, T)_j \neq y_\delta^*$. Assume $\max_j p(Y_\delta^* = j|x_\delta) \geq 1/2$, then there exists an interpolation ratio $\lambda$, such that*

$$\|\lambda \cdot f(x_\delta; \theta, T) + (1 - \lambda) \cdot p(\tilde{y}_\delta|x_\delta) - p(y_\delta^*|x_\delta)\|_{TV} \leq \|f(x_\delta; \theta, T) - p(y_\delta^*|x_\delta)\|_{TV}.$$

Note the above theorem focus on incorrectly classified examples and thus can be regarded as a complement to Theorem 4.1.

As a summarization, to reduce the distribution mismatch, we propose to use the following distribution as the assigned label of the adversarial example in adversarial training instead of the traditional adversarial label.

$$p_{\theta;T,\lambda}(y_\delta|x_\delta) = \lambda \cdot f(x_\delta; \theta, T) + (1 - \lambda) \cdot p(\tilde{y}_\delta|x_\delta), \tag{8}$$

We refer this label distribution as the *Rectified model probability*.

In Appendix D, we show that the optimal hyper-parameters (i.e. $T$ and $\lambda$) of almost all training examples concentrate on the same set of values by empirically studying on a synthetic dataset with known true label distribution. Therefore it is possible to find an universal set of hyper-parameters that reduce the distribution mismatch for all adversarial examples.

## 4.2 DETERMINE THE OPTIMAL TEMPERATURE AND INTERPOLATION RATIO

The set of temperature and interpolation ratio in the rectified model probability that maximally reduces the distribution mismatch is not straightforward to find as the true label distribution of the adversarial example is unknown in reality. Fortunately, given a sufficiently large validation dataset as a whole, it is possible to measure the overall distribution mismatch in a frequentist's view without knowing the true label distribution of every single example. A popular metric adopted here is the negative log-likelihood (NLL) loss, which is known as a proper scoring rule (Gneiting & Raftery, 2007) and is also employed in the confidence calibration of deep networks (Guo et al., 2017). By Gibbs's inequality it is easy to show that the NLL loss will only be minimized when the assigned label distribution matches the true label distribution (Hastie et al., 2001), namely

$$-\mathbb{E}_{(x_\delta, y_\delta^*) \in \mathcal{D}_\delta^{\text{val}}} \log p_{\theta;T,\lambda}(Y_\delta = y_\delta^*|x_\delta) \geq -\mathbb{E}_{p(y_\delta^*) \sim \mathcal{D}_\delta^{\text{val}}} p(y_\delta^*|x_\delta) \log p(y_\delta^*|x_\delta). \tag{9}$$

And here $y_\delta^*$ is known since the adversarial example should share the same argmax label with its clean counterpart as shown in Section 3.2.

Therefore, we propose to find the optimal $T$ and $\lambda$ in Equation (8) as

$$T, \lambda = \arg\min_{T,\lambda} -\mathbb{E}_{(x_\delta, y_\delta^*) \in \mathcal{D}_\delta^{\text{val}}} \log p_{\theta;T,\lambda}(Y_\delta = y_\delta^*|x_\delta) \tag{10}$$

One may note that Equation (10) cannot be directly optimized since the traditional adversarial label is only defined on the example in the training set and cannot be simply generalized to the validation set. A reasonable solution is using the nearest neighbour classifier to find the closest traditional adversarial label for every example in the validation set. However, to speed up the optimization we propose to employ the classifier overfitted by the traditional adversarial labels on the training set as an surrogate (see Appendix E.2), which works well in practice. Such process incurs almost no additional computation as we simply obtain the logits of a surrogate classifier on the validation set.

### 4.3 RECTIFIED MODEL PROBABILITY MITIGATES DOUBLE DESCENT

We now work on a realistic dataset (CIFAR-10) to demonstrate the rectified model probability proposed in Equation (8) can effectively mitigate the robust overfitting, or equivalently the epoch-wise double descent in adversarial training. The outer minimization of adversarial training (Equation (1)) now becomes

$$\theta^* = \arg\min_{\theta} \mathbb{E}_{\mathcal{D}_\delta} \, \ell\left(f(x_\delta; \theta), p_{\theta^{\text{Trad}}; T, \lambda}(y_\delta | x_\delta)\right), \tag{11}$$

where $\theta^{\text{Trad}}$ denotes the parameters of a classifier trained on the traditional adversarial label beforehand. The details of the experimental setting are available in Appendix F.1.

As shown in Figure 4, adversarial training on rectified model probability can best mitigate the robust overfitting when the temperature $T$ and interpolation ratio $\lambda$ are optimal. Such optimal hyperparameters perfectly aligns with the ones automatically determined by Equation (10).

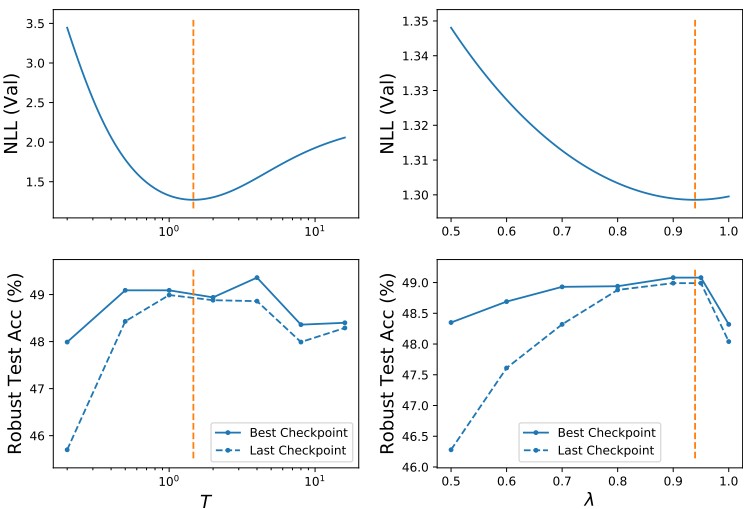

Figure 4: (Upper) NLL loss obtained on the validation set for different $T$ and $\lambda$. (Bottom) Robust test accuracy at the best and last checkpoint by adversarial training with the rectified model probability with different $T$ and $\lambda$. $\lambda = 0.8$ for grid search on $T$ (Left) and $T = 2$ for grid search on $\lambda$ (Right). Orange dashed lines in the left and right columns indicate the temperature and the interpolation ratio automatically determined by Equation (10), respectively.

## 5 EXPERIMENTS

**Experiment setup.** We consider three datasets, CIFAR-10, CIFAR-100 (Krizhevsky, 2009) and Tiny-ImageNet (Le & Yang, 2015). We consider robustness against $\ell_\infty$ norm-bounded adversarial attack with perturbation radius $8/255$, and employ AutoAttack (Croce & Hein, 2020) for reliable evaluation. We conduct PGD training on pre-activation ResNet-18 (He et al., 2016) in this section. Appendix C.1 includes results on additional adversarial training methods (e.g., TRADES (Zhang et al., 2019), FGSM (Goodfellow et al., 2015)), neural architectures (e.g., VGG (Simonyan & Zisserman, 2015), WRN) and evaluation metrics (e.g., PGD-1000, Square Attack (Andriushchenko et al., 2020), RayS (Chen & Gu, 2020)). More setup details can be found in Appendix F.

**Results & Discussions.** Our method is essentially adversarial training with self-distillation equipped with an algorithm automatically searching for the optimal hyper-parameters, which we now denote as KD-AT-Auto. We compare KD-AT-Auto with two baselines: regular adversarial training on traditional adversarial label (AT), and adversarial training combined with self-distillation (KD-AT) with fixed temperature $T = 2$ and interpolation ratio $\lambda = 0.5$ as suggested by Chen et al. (2021).

As shown in Figure 5, our method can effectively mitigate robust overfitting for all datasets, with both standard accuracy (SA) and robust accuracy (RA) constantly increasing throughout training. In Table 1, we measure the difference between the RA at the best checkpoint (Best) and at the last

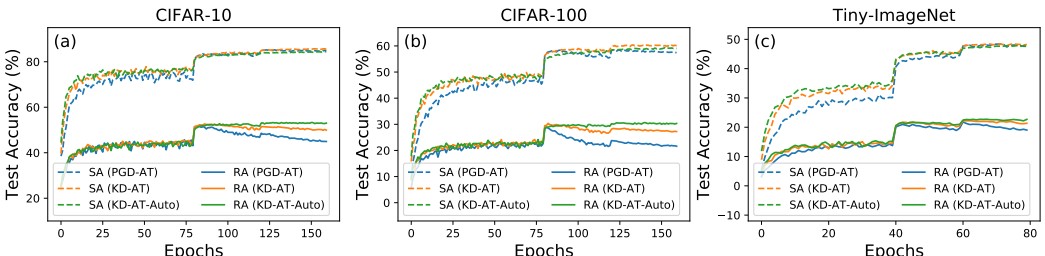

Figure 5: Our method can effectively mitigate robust overfitting for different datasets.

checkpoint (Last) to clearly show the overfitting gap. Our method can reduce the overfitting gap to less than $0.3\%$ for all datasets. One may note that self-distillation with fixed hyper-parameters is in fact inferior in terms of reducing robust overfitting, while its effectiveness can be significantly improved with the optimal hyper-parameters automatically determined by our method, which further verifies our understanding of robust overfitting.

Compared with self-distillation with fixed hyper-parameters, our method can also boost both RA and SA at the best checkpoint for all datasets.

Our method can further be combined with orthogonal techniques such as Stochastic Weight Averaging (SWA) (Izmailov et al., 2018) and additional standard teachers as mentioned in previous work (Chen et al., 2021) to achieve better performance. More results and discussion can be found in Appendix C.2.

| Dataset | Setting | $T$ | $\lambda$ | Robust Acc. (%) | | | Standard Acc. (%) | | |
|---|---|---|---|---|---|---|---|---|---|
| | | | | Best | Last | Diff. | Best | Last | Diff. |
| CIFAR-10 | AT | - | - | 47.35 | 41.42 | 5.93 | 82.67 | 84.91 | -2.24 |
| | KD-AT | 2 | 0.5 | 48.76 | 46.33 | 2.43 | 82.89 | **85.49** | -2.60 |
| | KD-AT-Auto | 1.47* | 0.8* | **49.05** | **48.80** | **0.25** | 84.26 | 84.47 | **-0.21** |
| CIFAR-100 | AT | - | - | 24.79 | 19.75 | 5.04 | 57.33 | 57.42 | -0.09 |
| | KD-AT | 2 | 0.5 | 25.77 | 23.58 | 2.19 | 57.24 | **60.04** | -2.80 |
| | KD-AT-Auto | 1.53* | 0.83* | **26.36** | **26.24** | **0.12** | 58.80 | 59.05 | **-0.25** |
| Tiny-ImageNet | AT | - | - | 17.20 | 15.40 | 1.80 | 47.72 | 47.62 | 0.10 |
| | KD-AT | 2 | 0.5 | 17.86 | 17.18 | 0.68 | **47.73** | **48.28** | -0.55 |
| | KD-AT-Auto | 1.23* | 0.85* | **18.29** | **18.39** | **-0.10** | 47.46 | 47.56 | **-0.10** |

Table 1: Performance of our method on different datasets. * denotes the hyper-parameter automatically determined by our method.

## 6    CONCLUSION AND DISCUSSIONS

In this paper, we extend the understanding of robust overfitting and show that it is the early part of an epoch-wise double descent in adversarial training. Our further analyses show that the double descent may originate from the implicit label noise introduced by the improper labeling of the adversarial examples. Based on our understanding, we propose an alternative labeling of adversarial examples by rectifying model probability, which can effectively mitigate robust overfitting without any manual hyper-parameter tuning.

By showing robust overfitting and epoch-wise double descent are in fact the same phenomena, our work eases the difficulty in understanding overfitting in modern generalization theory. Existing theories on double descent can be readily applied to adversarially robust learning setting, and potentially consolidate the theoretical grounding of robust learning.

Beyond the double descent in adversarial training, our definition of implicit label noise from a distribution mismatch perspective may also exist in a variety of scenarios, especially in real-world tasks where labeling is often complicated and ambiguous. Our method to rectify such label noise should thus be applicable as well. This would be an interesting future work.

REPRODUCIBLITY STATEMENT

We conduct experiments on public benchmark. We will release implementations for all methods and scripts for all experiments on GitHub, under the Apache-2.0 license.

ETHIC STATEMENT

In this paper, we explore the double descent phenomenon in adversarial robust learning from the implicit label noise perspective. We experiment on three benchmark datasets that are publicly available. Our analyses offer more in-depth understandings about adversarial training and can better improve the robustness of neural network models. Therefore, we believe our work is ethically on the right side of spectrum and has no potential for misuse, and cannot harm any vulnerable population.

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

# A PROOFS

## A.1 PROOFS AND REMARKS FOR IMPLICIT LABEL NOISE

**Total variation distance.**

For two discrete probability distributions $p(y)$ and $p(y')$ where $y, y' \in \mathcal{Y}$, the total variation distance between them can be equally defined as

$$\|p(y) - p(y')\|_{\mathrm{TV}} = \max_J \left| \sum_{j \in J} p(y = j) - \sum_{j \in J} p(y' = j) \right|$$

$$= \frac{1}{2} \sum_j |p(y = j) - p(y' = j)|$$

**Proof of Theorem 3.1.** This theorem can be easily proven by the *coupling inequality*.

**Proof of Theorem 3.2.**

*Proof.* For simplicity, we consider the adversarial perturbation generated by FGSM. Other adversarial perturbation can be viewed as a Taylor series of such perturbation.

$$\delta^* \approx -\varepsilon \frac{\nabla_x \, \max_j p(y^* = j|x)}{\|\nabla_x \, \max_j p(y^* = j|x)\|}, \tag{12}$$

Now let $j^* = \arg\max \, p(y = j|x)$. Recall $f^*(x) = p(y^*|x)$, $f^*(x_\delta) = p(y_\delta^*|x_\delta)$ and $p(y_\delta|x_\delta) = p(y^*|x)$. We have

$$\min p(y_\delta \neq y_\delta^*|x_\delta) = \|p(y^*|x) - p(y_\delta^*|x_\delta)\|_{\mathrm{TV}}$$

$$= \frac{1}{2} \sum_j |p(y^* = j|x) - p(y_\delta^* = j|x_\delta)| \quad \boxed{\text{TV distance}}$$

$$\geq \frac{1}{2} |p(y^* = j^*|x) - p(y_\delta^* = j^*|x_\delta)|$$

$$= \frac{1}{2} |f^*(x)_{j^*} - f^*(x + \delta)_{j^*}|$$

$$= \frac{1}{2} [f^*(x)_{j^*} - f^*(x + \delta)_{j^*}] \quad \boxed{\text{Adversarial perturbation}}$$

$$= \frac{1}{2} \left[ -\nabla_x f^*(x)_{j^*} \cdot \delta - \frac{1}{2} \delta^T \nabla^2 f^*(z)_{j^*} \delta \right]$$

$$\geq \frac{1}{2} \left[ -\nabla_x f^*(x)_{j^*} \cdot \delta - \frac{M}{2} \|\delta\|_2^2 \right] \quad \boxed{\text{Local convexity}}$$

$$\geq \frac{1}{2} \left[ \varepsilon \frac{\|\nabla_x f^*(x)_{j^*}\|_2^2}{\|\nabla_x f^*(x)_{j^*}\|} - \frac{M}{2} \|\delta\|_2^2 \right]$$

$$\geq \frac{1}{2} \left[ \varepsilon \|\nabla_x f^*(x)_{j^*}\|_2 - \frac{M}{2} \|\delta\|_2^2 \right]$$

$$\geq \frac{1}{2} \left[ \varepsilon \|\nabla_x f^*(x)_{j^*}\|_\infty - \frac{M}{2} \|\delta\|_2^2 \right],$$

where the last two inequalities leverage the fact that $\|\cdot\|_\infty \leq \|\cdot\|_2$.

With the assumption

$$\|\nabla_x \, f^*(x)_j\| \propto \begin{cases} 1 - f^*(x)_j, & f^*(x)_j \to 1 \\ f^*(x)_j, & f^*(x)_j \to 0, \end{cases}$$

we then have

$$\underline{\min} \, p(\hat{y} \neq y | x + \delta) \propto \varepsilon(1 - p(y = j^* | x)),$$

where $\underline{\min}$ means the lower bound of the minimum label noise.

$\square$

**Sanity check of Assumption 7.**    Here we check if Assumption 7 is reasonable by studying a Gaussian mixture model (GMM).

The conditional distribution in a Gaussian mixture model can be formulated as

$$p(y = j | x) \equiv f^*(x)_j = \frac{\psi_j \mathcal{N}_x(\mu_j, \sigma_j)}{\sum_l \psi_l \mathcal{N}_x(\mu_l, \sigma_l)} \equiv \frac{\psi_j g_j(x)}{\sum_l \psi_l g_l(x)},$$

where

$$g_l(x) = \frac{1}{\sqrt{\det(2\pi\sigma_l)}} \exp\left[-\frac{1}{2}(x - \mu_l)^T \sigma_l^{-1}(x - \mu_l)\right].$$

The gradient can be derived as

$$
\begin{aligned}
\nabla_x f^*(x)_j &= \frac{\psi_j \nabla g_j(x)}{\sum_l \psi_l g_l(x)} - \psi_j g_j(x) \frac{\sum_l \psi_l \nabla g_l(x)}{[\sum_l \psi_l g_l(x)]^2} \\
&= \frac{-\psi_j g_j(x) \sigma_j^{-1}(x - \mu_j)}{\sum_l \psi_l g_l(x)} + \psi_j g_j(x) \frac{\sum_l \psi_l g_l(x) \sigma_l^{-1}(x - \mu_l)}{[\sum_l \psi_l g_l(x)]^2} \\
&= \frac{\psi_j g_j(x)}{\sum_l \psi_l g_l(x)} \left[-\sigma_j^{-1}(x - \mu_j) + \frac{\sum_l \sigma_l^{-1}(x - \mu_l)\psi_l g_l(x)}{\sum_l \psi_l g_l(x)}\right] \\
&= f^*(x)_j \left[-\sigma_j^{-1}(x - \mu_j) + \sum_l f^*(x)_l \sigma_l^{-1}(x - \mu_l)\right]
\end{aligned}
$$

When $x - \mu_j \to 0$, $f^*(x)_j \to 1$ and $f^*(x)_l \to 0$, for $l \neq j$,

$$\nabla_x f^*(x)_j \approx f^*(x)_j (f^*(x)_j - 1)\sigma_j^{-1}(x - \mu_j)$$

Therefore

$$\|\nabla_x f^*(x)_j\| \propto 1 - f^*(x)_j$$

### A.2    PROOFS FOR MITIGATING DOUBLE DESCENT

**Proof of Theorem 4.1.**

*Proof.* Let $j^* = \arg\max p(y_\delta = j | x_\delta)$ and thus $p(y_\delta = j^* | x_\delta) \in [1/c, 1]$. Let $g(T) := f(x_\delta; \theta, T)_{j^*}$, which is a continuous function defined on $[0, \infty]$. The condition $j^* = \arg\max_j f(x_\delta; \theta, T)_j$ ensures that $g(T) \in [1/c, 1]$, where $c$ is the number of classes. By the intermediate value theorem, there exists $T^*$, such that $g(T^*) = p(y_\delta = j^* | x_\delta)$.

Let $T = T^*$, we have

$$
\begin{aligned}
\|f(x_\delta; \theta, T) - p(y_\delta^* | x_\delta)\|_{TV} &= \frac{1}{2} \sum_j |f(x_\delta; \theta, T)_j - p(y_\delta^* = j | x_\delta)| \\
&= \frac{1}{2} \sum_{j, j \neq j^*} |f(x_\delta; \theta, T)_j - p(y_\delta^* = j | x_\delta)| \\
&\leq \frac{1}{2} \left[\sum_{j, j \neq j^*} f(x_\delta; \theta, T)_j + \sum_{j, j \neq j^*} p(y_\delta^* = j | x_\delta)\right] \\
&= 1 - p(y_\delta = j^* | x_\delta),
\end{aligned}
$$

where the inequality holds by the triangle inequality.

Meanwhile, we have

$$
\begin{aligned}
\|p(\tilde{y}_\delta|x_\delta) - p(y_\delta^*|x_\delta)\|_{TV} &= \|p(y|x) - p(y_\delta^*|x_\delta)\|_{TV} \\
&= \|\mathbb{1}(y) - p(y_\delta^*|x_\delta)\|_{TV} \\
&= \frac{1}{2}\left[1 - p(y_\delta^* = y|x_\delta) + \sum_{j,j\neq\hat{y}} p(y_\delta^* = y|x_\delta)\right] \\
&= 1 - p(y_\delta^* = y|x_\delta) \\
&\geq 1 - p(y_\delta^* = j^*|x_\delta).
\end{aligned}
$$

Therefore, it can seen that for $T = T^*$,

$$
\|f(x_\delta;\theta,T) - p(y_\delta^*|x_\delta)\|_{TV} \leq \|p(\tilde{y}_\delta|x_\delta) - p(y_\delta^*|x_\delta)\|_{TV}.
$$

$\square$

**Proof of Theorem 4.2.**

**Lemma A.1.** *Let $x_\delta$ be an example incorrectly classified by a classifier $f$ in terms of the true label distribution $p(y_\delta^* = j|x_\delta)$, namely*

$$
\arg\max_j\ f(x_\delta;\theta,T)_j \neq j^*,
$$

*where $j^* = \arg\max_j\ p(y_\delta^* = j|x_\delta)$. Assume $p(y_\delta^* = j^*|x_\delta) \geq 1/2$, then*

$$
f(x_\delta;\theta,T)_{j^*} \leq p(y_\delta^* = j^*|x_\delta).
$$

*Proof.* We prove it by contradiction. Assume $f(x_\delta;\theta,T)_{j^*} > p(y_\delta^* = j^*|x_\delta)$, we have $f(x_\delta;\theta,T)_{j^*} > p(y_\delta^* = j^*|x_\delta) \geq 1/2$. Therefore,

$$
f(x_\delta;\theta,T)_j \leq \sum_{j,j\neq j^*} f(x_\delta;\theta,T)_j = 1 - f(x_\delta;\theta,T)_{j^*} < 1/2,\ \forall j \neq j^*,
$$

which means $f(x_\delta;\theta,T)_j < f(x_\delta;\theta,T)_{j^*},\ \forall j \neq j^*$. This leads to $j^* = \arg\max_j f(x_\delta;\theta,T)_j$, which contradicts our condition. $\square$

Now we prove Theorem 4.2

*Proof.* First let $p(y_\delta|x_\delta) = p(y|x) \approx \mathbb{1}(y)$. This is our assumption. But the approx here would be a problem, we need exactly one-hot.

Let $j^* = \arg\max_j p(y_\delta^* = j|x_\delta)$. By Lemma A.1 we have $f(x_\delta;\theta,T)_{j^*} \leq p(y_\delta^* = j^*|x_\delta) \leq 1$. Then there exists $\lambda^* > 0$, such that $\lambda^* \cdot f(x_\delta;\theta,T)_{j^*} + (1-\lambda^*) = p(y_\delta^* = j^*|x_\delta)$ by the intermediate value theorem.

Let $\lambda = \lambda^*$, we have

$$2\left[\|\lambda \cdot f(x_\delta; \theta, T) + (1-\lambda) \cdot p(\tilde{y}_\delta|x_\delta) - p(y_\delta^*|x_\delta)\|_{TV} - \|f(x_\delta; \theta, T) - p(y_\delta^*|x_\delta)\|_{TV}\right]$$

$$=2\left[\|\lambda \cdot f(x_\delta; \theta, T) + (1-\lambda) \cdot \mathbb{1}(y) - p(y_\delta^*|x_\delta)\|_{TV} - \|f(x_\delta; \theta, T) - p(y_\delta^*|x_\delta)\|_{TV}\right]$$

$$=\sum_j |\lambda \cdot f(x_\delta; \theta, T)_j + (1-\lambda) \cdot 1(j=y) - p(y_\delta^* = j|x_\delta)| - \sum_j |f(x_\delta; \theta, T)_j - p(y_\delta^* = j|x_\delta)|$$

$$=\sum_j |\lambda \cdot f(x_\delta; \theta, T)_j + (1-\lambda) \cdot 1(j=y^*) - p(y_\delta^* = j|x_\delta)| - \sum_j |f(x_\delta; \theta, T)_j - p(y_\delta^* = j|x_\delta)|$$

$$=\sum_{j,j\neq j^*} |\lambda \cdot f(x_\delta; \theta, T)_j - p(y_\delta^* = j|x_\delta)| - \sum_{j,j\neq j^*} |f(x_\delta; \theta, T)_j - p(y_\delta^* = j|x_\delta)| - |f(x_\delta; \theta, T)_{j^*} - p(y_\delta^* = j^*|x_\delta)|$$

$$\leq \sum_{j,j\neq j^*} |\lambda \cdot f(x_\delta; \theta, T)_j - f(x_\delta; \theta, T)_j| - |f(x_\delta; \theta, T)_{j^*} - p(y_\delta^* = j^*|x_\delta)|$$

$$=\sum_{j,j\neq j^*} [f(x_\delta; \theta, T)_j - \lambda \cdot f(x_\delta; \theta, T)_j] - [p(y_\delta^* = j^*|x_\delta) - f(x_\delta; \theta, T)_{j^*}]$$

$$=\sum_{j,j\neq j^*} [f(x_\delta; \theta, T)_j - \lambda \cdot f(x_\delta; \theta, T)_j] - [\lambda \cdot f(x_\delta; \theta, T)_{j^*} + (1-\lambda) - f(x_\delta; \theta, T)_{j^*}]$$

$$=\sum_j f(x_\delta; \theta, T)_j - \lambda \sum_j f(x_\delta; \theta, T)_j - (1-\lambda)$$

$$= 0.$$

$\square$

## B   MORE EMPIRICAL ANALYSES

### B.1   EPOCH-WISE DOUBLE DESCENT IS UBIQUITOUS IN ADVERSARIAL TRAINING

In this section, we conduct extensive experiments with different optimizers, sample sizes, model architectures, and learning rate schedulers to verify the connection between robust overfitting and epoch-wise double descent. The default experiment settings are listed in Appendix F.2 in detail.

**Optimizer.**   Similar to the setting employed in Nakkiran et al. (2020), we conduct the adversarial training using both the Adam optimizer and SGD. As already shown in Figure 1, double descent can be observed for both optimizers, although Adam may be inferior compared to SGD.

**Sample size.**   We randomly sample a desired number of examples without replacement from the original training set in CIFAR-10. As shown in Figure 6, for both optimizers, increasing sample size will shrink the area under the double descent curve, but will not significantly distort its shape. Similar observation is also made for double descent curve under standard training (Nakkiran et al., 2020).

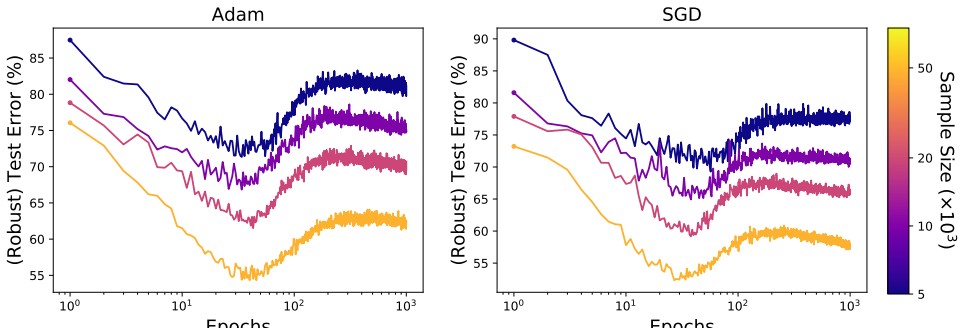

Figure 6: Varying sample size will shrink the area under the epoch-wise double descent curve, but will not significantly distort its shape.

**Model capacity.**   We modulate the capacity of the deep model by varying the widening factor of the Wide ResNet. To extend the lower limit of the capacity, we allow the widening factor to be less

than 1, in which case the number of channels in each residual block is scaled similarly but rounded. The number of channels in the first convolutional layer will be reduced accordingly to ensure the width monotonically increasing through the forward propagation. To accelerate the training with an extremely large model, we randomly sample a training set of size 5000 and employ the Adam optimizer, since the sample size will not significantly distort the shape of the double descent as shown above. Figure 7a shows that the double descent will gradually become more complete as the model capacity increases and the model translates from under-parameterized to over-parameterized regime (Nakkiran et al., 2020).

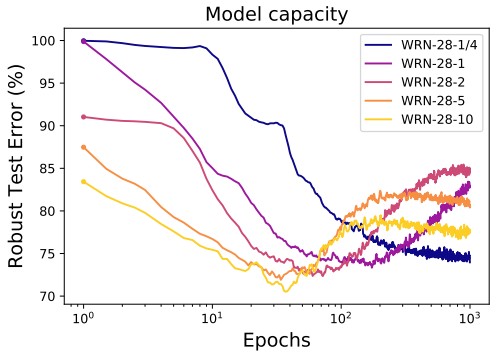

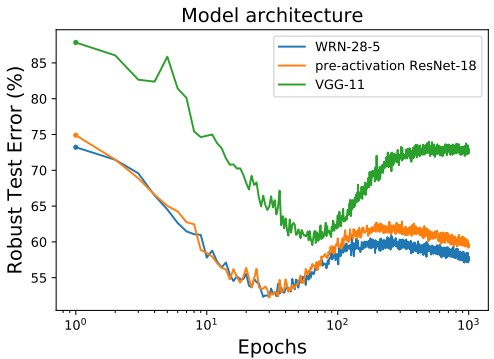

(a) Similar to Figure 1 in the introduction, but include more widening factors to show the gradual transition of the model from under-parameterized to over-parameterized regime. The training curves are smoothed by a window of 5.

(b) Different model architectures will affect the double descent curve. In particular, VGG-11 will have the second descent delayed due to its inferior performance compared to residual architectures.

Figure 7: Effect of model on the epoch-wise double descent curve

**Model architecture.** We also experiment on model architectures other than Wide ResNet, including pre-activation ResNet-18 (He et al., 2016) and VGG-11 (Simonyan & Zisserman, 2015). We select these configurations to ensure approximately comparable model capacities[2]. As shown in Figure 7, different model architectures may produce slightly different double descent curves. The second descent of VGG-11 in particular will be delayed due to its inferior performance compared to residual architectures.

**Learning rate scheduler.** A specific learning rate scheduler may shape the robust overfitting differently as suggested by Rice et al. (2020). We consider the following learning rate schedulers in our experiments.

- **Piecewise decay**: The initial learning rate rate is set as 0.1 and is decayed by a factor of 10 at the 100th and 500th epochs within a total of 1000 epochs.
- **Cyclic**: This scheduler was initially proposed by Smith (2017) and has been popular in adversarial training. We set the maximum learning rate to be 0.2, and the learning rate will linearly increase from 0 to 0.2 for the initial 400 epochs and decrease to 0 for the later 600 epochs.
- **Cosine**: This scheduler was initially proposed by Loshchilov & Hutter (2017). The learning rate starts at 0.1 and gradually decrease to 0 following a cosine function for a total of 1000 epochs.

Experiments on various learning rate schedulers show the second descent can be widely observed except the piecewise decay, where the appearance of second descent might be delayed due to extremely small learning rate in the late stage of training. This further demonstrates the connection between robust overfitting and epoch-wise double descent.

## B.2 DEPENDENCE OF DOUBLE DESCENT IN ADVERSARIAL TRAINING

In this section, we show that the double descent in adversarial training is conditioned on the data quality. As models are trained on adversarial examples, data properties in adversarial training are

---

[2]WRN-28-5, pre-activation ResNet-18 and VGG-11 have $9.13 \times 10^6$, $11.17 \times 10^6$ and $9.23 \times 10^6$ parameters, respectively.

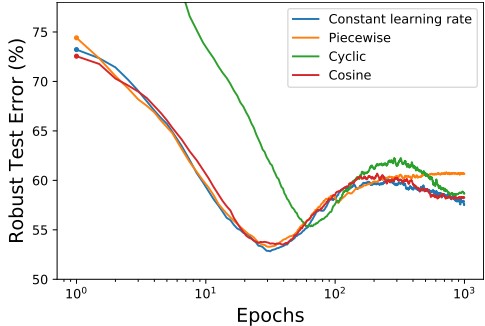

Figure 8: The effect of the learning rate scheduler on the epoch-wise double descent curve in adversarial training. Modulating the model capacity can produce training curves with diverse behaviors. Different model architectures may produce slightly different double descent curves. The training curve is smoothed by moving average with a window of $5$.

concerned with the quality of the clean examples, and the adversary employed to generate the perturbation, which further depends on the perturbation radius and the number of attack iterations[3]. We show that the double descent in adversarial training has a strong dependence on the data quality and the perturbation radius, but almost no dependence on the number of attack iterations.

**Perturbation radius.** Overfitting has been shown to dominate in adversarial training, but rarely appear in standard training (Rice et al., 2020). This suggests the overfitting, or more generally double descent, is conditioned on the perturbation radius in adversarial training, since standard training and standard accuracy can be viewed as adversarial training and robust accuracy with zero perturbation radius, respectively. By modulating the perturbation radius, we show that such correlation is gradual.

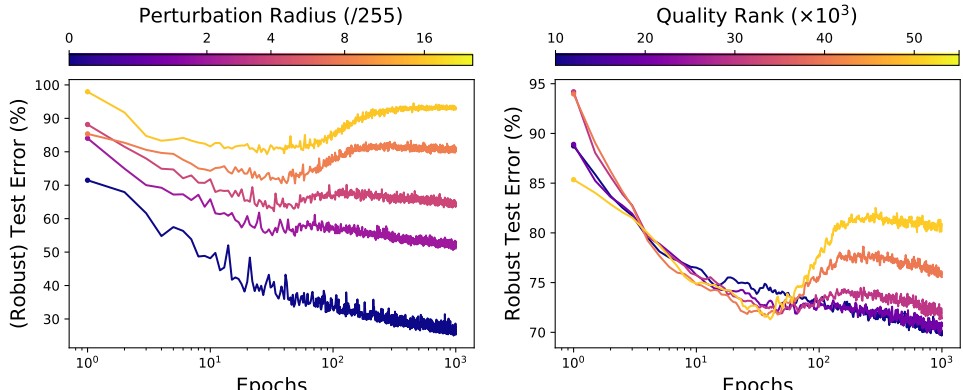

Figure 9: (Left) Dependence of double descent on the perturbation radius. $\varepsilon = 0/255$ indicates the standard training where no double descent occurs. (Right) Dependence of double descent on the data quality. The curves are smoothed by a window of $5$ epochs to reduce overlapping.

As shown in Figure 9, the double descent emerges and exacerbates as the perturbation radius increases, indicating a strong correlation between perturbation radius and double descent in adversarial training. In particular, when the perturbation radius is around $4/255$, a second minimum can be observed which is equivalently good or even better than the first minimum. This again validates the strong connection between robust overfitting and double descent, and suggests longer training can still be helpful for adversarial training.

**Data quality.** Previous works have shown that some datasets such as ImageNet (Deng et al., 2009) may produce significantly stronger robust overfitting (Rice et al., 2020). It has also been observed that the low-quality data in the same dataset causes the robust overfitting (Dong et al., 2021a). This

---

[3]The best step size in terms of the model performance can often be determined from the combination of perturbation radius and attack iterations (Madry et al., 2018)

implies that the double descent in adversarial training hinges on the quality of the data. We measure the data quality using the predictive probability of a classifier (see Appendix B.2 for details), and sample training sets with different levels of data quality controlled by a threshold of the quality-based rank. Figure 9 shows that as the quality of the training set degrades, the double descent gradually emerges and exacerbates, indicating a strong correlation between the data quality and the double descent in adversarial training. One may again note that when the quality of the training data is relatively high, a second minimum can be observed which is equivalently good as the first minimum.

**The number of attack iterations.** We have shown that the double descent in adversarial training strongly depends on the perturbation radius. In this section we conduct experiments to show whether it also depends on the strength of the adversary.

In Figure 10, we fix the perturbation radius as $4/255$ where the double descent is relatively complete and vary the number of attack iterations of the PGD attack employed in the inner maximization. One may find that as long as the model capacity is reasonably large, the number of attack iterations will not significantly affect the double descent curve, both for epoch-wise one and model-wise one. From the analysis of implicit label noise, this is easy to understand as more attack iterations will not reduce the probability corresponding to the true label much more—it is widely observed more iterations in PGD attack only marginally increase the attack successful rate. Consequently, the distribution mismatch between the true label distribution and the assigned label distribution that induces the implicit label noise will not expand significantly.

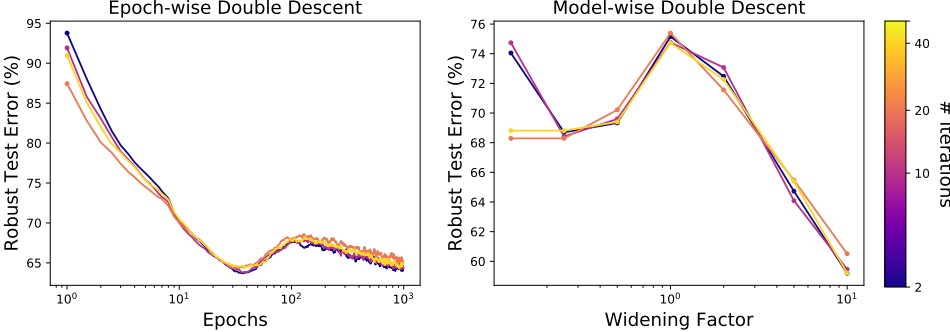

Figure 10: Dependence of double descent on the number of attack iterations. As more iterations are employed in the inner maximization, neither epoch-wise nor model-wise double descent changes significantly except for an extremely small model (WRN-28-1/8). For the model-wise double descent, only the test error at the best checkpoint is shown to avoid overlapped curves since the last checkpoint achieves similar error as the best checkpoint in this training setting.

### B.3 GAUSSIAN NOISE

One may expect adversarial training and adversarial augmentation (see Section 3.5) produce double descent simply by perturbing the inputs, thus increasing the variance. To verify the implicit label noise induced by a mismatch between the true label distribution and the assigned label distribution is essential to produce double descent, we select one type of common corruption, Gaussian noise, to perturb the image inputs. We first briefly show that Gaussian noise does not cause a distribution mismatch.

We follow the notation introduced in Section 3.1, where $\delta \in \mathbb{R}^d$ now refers to a Gaussian perturbation. Under a 1-st order approximation of $p(y|x)$, the mismatch between the true label distribution and label distribution can be derived as (see proof of Theorem 3.2 in Appendix A.1)

$$\|p(y_\delta^*|x_\delta) - p(\tilde{y}_\delta|x_\delta)\|_{\text{TV}} \equiv \|p(y_\delta^*|x_\delta) - p(y|x)\|_{\text{TV}} = \frac{1}{2}\sum_j \left|\nabla_x f^*(x)_j \cdot \delta + \frac{1}{2}\delta^T \nabla^2 f^*(z)_j \delta\right|$$

(13)

where $\delta \sim \mathcal{N}(0, \sigma \cdot I_d)$. Since the image inputs span a low-dimension subspace $\mathcal{X}$ and the dimensionality $d$ is large (3072 for CIFAR-10), it is highly likely that $\delta \perp \mathcal{X}$, which means $\nabla_x f^*(x)_j \cdot \delta = 0$ and $\delta^T \nabla^2 f^*(z)_j \delta = 0$ for all $j$. One can also empirically verify that a Gaussian perturbation is almost always orthogonal to the difference between any two images, while an adversarial perturbation is not.

We now experiment on dataset perturbed by Gaussian noise and verify our intuition. Similar to adversarial augmentation, we apply Gaussian noise to the training set only once, and then conduct standard training on the perturbed training set. Aligned with the setting of adversarial augmentation (See Appendix F.3), we employ Adam to train a WRN-28-5 on randomly selected 5000 examples for 1000 epochs. As shown in Figure 11, Gaussian noise with a perturbation radius as high as $80/255$, which will reduce the accuracy of a classifier to the same level as an adversarial attack will, does not produce significant double descent. We note that similar observation has been made on common curruption benchmark CIFAR-10-C (Hendrycks & Dietterich, 2019) in a previous work (Yang et al., 2020b).

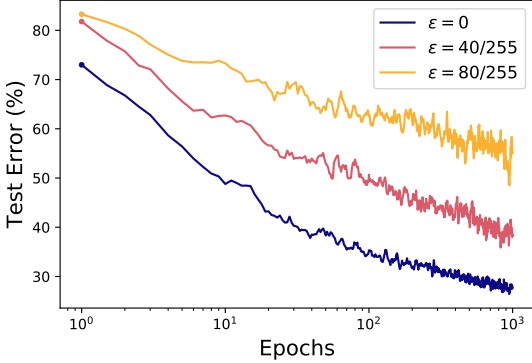

Figure 11: Even with extremely high Gaussian noise corrupting the training set, no significant double descent can be observed. This shows the input perturbation is not essential to produce double descent.

## C MORE EXPERIMENT RESULTS

### C.1 ADVERSARIAL TRAINING METHODS, NEURAL ARCHITECTURES AND EVALUATION METRICS

In this section we conduct extensive experiments with different adversarial training methods, neural architectures and robustness evaluation metrics to verify the effectiveness of our method.

| Method | Setting | $T$ | $\lambda$ | Robust Acc. (%) | | | Standard Acc. (%) | | |
|---|---|---|---|---|---|---|---|---|---|
| | | | | Best | Last | Diff. | Best | Last | Diff. |
| TRADES | AT | - | - | 48.50 | 45.53 | 2.97 | 82.79 | 82.68 | 0.11 |
| | KD-AT | 2 | 0.5 | 48.74 | 47.52 | 1.22 | 82.30 | **83.03** | -0.73 |
| | KD-AT-Auto | $1.12^*$ | $0.82^*$ | **48.75** | **48.39** | **0.36** | **82.44** | 82.80 | **-0.36** |
| FGSM | AT | - | - | 41.96 | 35.39 | 6.57 | 85.91 | 87.20 | -1.29 |
| | KD-AT | 2 | 0.5 | 42.82 | 41.61 | 1.21 | 86.69 | **87.93** | -1.24 |
| | KD-AT-Auto | $2.18^*$ | $0.78^*$ | **44.11** | **43.75** | **0.36** | **87.38** | 87.66 | **-0.28** |

Table 2: Performance of our method with different adversarial training methods.

| Architecture | Setting | $T$ | $\lambda$ | Robust Acc. (%) | | | Standard Acc. (%) | | |
| --- | --- | --- | --- | --- | --- | --- | --- | --- | --- |
| | | | | Best | Last | Diff. | Best | Last | Diff. |
| VGG-19 | AT | - | - | 42.21 | 39.12 | 3.09 | 73.95 | 80.45 | -6.50 |
| | KD-AT | 2 | 0.5 | 43.59 | 42.69 | 0.90 | 74.30 | **77.80** | -3.50 |
| | KD-AT-Auto | 1.28* | 0.79* | **44.27** | **44.24** | **0.03** | 76.41 | 76.79 | **-0.38** |
| WRN-28-5 | AT | - | - | 49.85 | 42.89 | 6.96 | 84.82 | 85.87 | -1.05 |
| | KD-AT | 2 | 0.5 | 51.08 | 48.40 | 2.68 | 85.36 | **86.88** | -1.52 |
| | KD-AT-Auto | 1.6* | 0.82* | **51.47** | **51.10** | **0.37** | 86.05 | 86.24 | **-0.19** |
| WRN-34-10 | AT | - | - | 52.29 | 46.04 | 6.25 | 86.57 | 86.75 | -0.18 |
| | KD-AT | 2 | 0.5 | 53.11 | 50.97 | 2.14 | 86.41 | **88.06** | -1.65 |
| | KD-AT-Auto | 1.6* | 0.83* | **54.17** | **53.71** | **0.46** | 87.69 | 88.01 | **-0.32** |

Table 3: Performance of our method with different neural architectures.

| Attacks | Setting | $T$ | $\lambda$ | Robust Acc. (%) | | |
| --- | --- | --- | --- | --- | --- | --- |
| | | | | Best | Last | Diff. |
| PGD-1000 | AT | - | - | 50.64 | 43.00 | 7.64 |
| | KD-AT | 2 | 0.5 | 51.79 | 48.43 | 3.36 |
| | KD-AT-Auto | 1.47* | 0.8* | **52.05** | **51.71** | **0.34** |
| Square Attack | AT | - | - | 53.47 | 48.90 | 4.57 |
| | KD-AT | 2 | 0.5 | 54.39 | 52.92 | 1.47 |
| | KD-AT-Auto | 1.28* | 0.79* | **55.23** | **55.17** | **0.06** |
| RayS | AT | - | - | 55.76 | 51.63 | 4.13 |
| | KD-AT | 2 | 0.5 | 56.59 | 55.50 | 1.09 |
| | KD-AT-Auto | 1.6* | 0.82* | **57.74** | **57.54** | **0.20** |

Table 4: Performance of our method under different adversarial attacks. PGD-1000 refers to PGD attack with 1000 attack iterations, with step size fixed as $2/255$ as recommended by Croce & Hein (2020).

## C.2 COMBINED WITH ADDITIONAL TECHNIQUES

Here, we show that combined with the additional techniques proposed in (Chen et al., 2021), our method can achieve better performance.

We note that our proposed method is essentially the baseline knowledge distillation for adversarial training with a robustly trained self-teacher, equipped with an algorithm that automatically finds its optimal hyperparameters (i.e. the temperature $T$ and the interpolation ratio $\lambda$). Stochastic Weight Averaging (SWA) and additional standard teachers employed in (Chen et al., 2021) are orthogonal contributions. KD-AT-Auto can certainly be combined with SWA and KD-Std to achieve better performance. As shown in Table 5, on CIFAR-10, KD-AT + KD-Std + SWA (Chen et al., 2021) can already reduce the overfitting gap (difference between the best and last robust accuracy) to almost 0, while KD-AT-Auto + KD-Std + SWA maintains an overfitting gap close to 0. Interestingly, on the SVHN dataset (Netzer et al., 2011), where KD-AT + KD-Std + SWA still produces a high overfitting gap (also see Appendix A1.3 in (Chen et al., 2021)), KD-AT-Auto + KD-Std + SWA can further push this gap to close to 0.

Here, the interpolation ratio of the standard teacher is fixed as 0.2 and the SWA starts at the first learning rate decay for all experiments. We employ PGD-AT (Madry et al., 2018) as the base adversarial training method and conduct experiments with a pre-activation ResNet-18. The robust accuracy is evaluated with AutoAttack. Other experiment details are in line with Appendix F.1.

Furthermore, we note that (Chen et al., 2021) shows SWA and KD-Std are essential components to mitigate robust overfitting on top of KD-AT, while we show that KD-AT itself can mitigate robust overfitting by proper parameter tuning. We are thus able to separate these components and allow a more flexible selection of hyperparameters in diverse training scenarios without fear of overfitting. In particular, although (Chen et al., 2021) suggests SWA starting at the first learning rate decay (exactly when the overfitting starts) mitigates robust overfitting, the effectiveness of SWA on mitigating

| Dataset | Setting | $T$ | $\lambda$ | Robust Acc. (%) | | | Standard Acc. (%) | | |
|---|---|---|---|---|---|---|---|---|---|
| | | | | Best | Last | Diff. | Best | Last | Diff. |
| CIFAR-10 | AT | - | - | 47.35 | 41.42 | 5.93 | 82.67 | 84.91 | -2.24 |
| | KD-AT + KD-Std + SWA | 2 | 0.5 | 49.98 | 49.89 | 0.09 | **85.06** | **85.52** | -0.46 |
| | KD-AT-Auto + KD-Std + SWA | 1.47* | 0.8* | **50.03** | **50.05** | **-0.02** | 84.69 | 84.91 | **-0.22** |
| SVHN | AT | - | - | 47.83 | 39.77 | 8.06 | 90.18 | 91.11 | -0.93 |
| | KD-AT + KD-Std + SWA | 2 | 0.5 | 47.88 | 46.46 | 1.42 | **91.59** | **91.76** | **-0.17** |
| | KD-AT-Auto + KD-Std + SWA | 1.53* | 0.83* | **50.58** | **50.09** | **0.49** | 90.54 | 90.76 | -0.22 |

Table 5: Performance of our method combined with SWA and additional standard teacher on different datasets.

overfitting may strongly depend on its hyper-parameter selection including $s_0$, i.e., the starting epoch and $\tau$, i.e., the decay rate[4], which is also mentioned in recent work (Rebuffi et al., 2021). We also did some additional experiments on CIFAR-10 following the SWA setting in (Rebuffi et al., 2021) to demonstrate the wide applicability of our method. As shown by Table 6, when changing the hyperparameters of SWA, KD-AT + KD-Std + SWA cannot consistently mitigate robust overfitting, while KD-AT-Auto + KD-Std + SWA can maintain an overfitting gap close to 0 and achieve better robustness as well.

| Setting | $s_0$ | $\tau$ | Robust Acc. (%) | | | Standard Acc. (%) | | |
|---|---|---|---|---|---|---|---|---|
| | | | Best | Last | Diff. | Best | Last | Diff. |
| KD-AT + KD-Std + SWA | 80 | 0.999 | 49.00 | 48.04 | 0.96 | 84.04 | **86.11** | -2.07 |
| KD-AT-Auto + KD-Std + SWA | 80 | 0.999 | **49.35** | **49.25** | **0.1** | **85.38** | 85.91 | **-0.37** |
| KD-AT + KD-Std + SWA | 0 | 0.999 | 49.01 | 48.01 | 1.0 | 83.78 | **86.20** | -2.42 |
| KD-AT-Auto + KD-Std + SWA | 0 | 0.999 | **49.32** | **49.25** | **0.07** | **84.78** | 85.48 | **-0.7** |

Table 6: Performance of our method combined with SWA with different hyper-parameters

# D STUDY ON A SYNTHETIC DATASET WITH KNOWN TRUE LABEL DISTRIBUTION

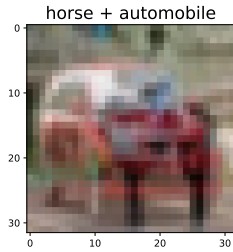

Figure 12: Example mixup augmentation.

**Synthetic Dataset.** Since the true label distribution is typically unknown for adversarial examples in real-world datasets, we simulate the mechanism of implicit label noise in adversarial training from a feature learning perspective. Specifically, we adapt *mixup* (Zhang et al., 2018) for data augmentation on CIFAR-10. For every example $x$ in the training set, we randomly select another example $x'$ in a different class and linearly interpolate them by a ratio $\rho$, namely $x := \rho x + (1 - \rho)x'$, which essentially perturbs $x$ with features from other classes. Therefore, the true label distribution is arguably $y \sim \rho \cdot \mathbb{1}(y) + (1-\rho) \cdot \mathbb{1}(y')$. Unlike mixup, we intentionally set the assigned label as $\hat{y} \sim \mathbb{1}(y)$, thus deliberately create a mismatch between the true label distribution and the assigned label distribution. We refer this strategy as *mixup augmentation* and only perform it once before the training. In this way, the true label distribution of every example in the synthetic dataset is fixed.

**Concentration of optimal temperature and interpolation ratio of individual examples.** In Section 4.1 we have shown that in terms of individual examples, the rectified model probability can provably reduce the distribution mismatch between the assigned label distribution and true label distribution of the adversarial example. However, since the true label distribution is unknown in realistic scenarios, it is not possible to directly follow Theorems 4.1 and 4.2 and calculate the optimal set of hyper-parameters for each example in the training set. The best we can do is to employ a

---

[4]SWA can be implemented using an exponential moving average $\theta'$ of the model parameters $\theta$ with a decay rate $\tau$, namely $\theta' \leftarrow \tau \cdot \theta' + (1 - \tau) \cdot \theta$ at each training step (Rebuffi et al., 2021).

validation set and determine a universal set of hyper-parameters based on the NLL loss, which expects all training examples to share similar optimal temperatures and interpolation ratios. Here, based on the synthetic dataset where a true label distribution is known, we empirically verify this assumption is reasonable.

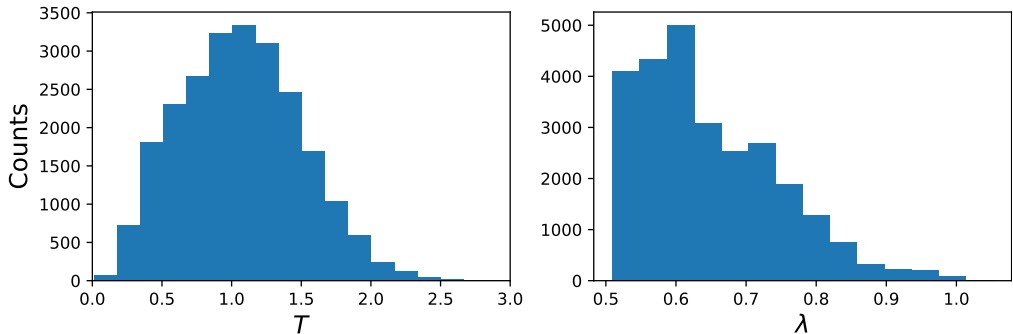

Figure 13: The histograms of optimal temperature (left) and interpolation ratio (right) of individual examples.

In Figure 13 left, we solve the optimal temperature for each correctly classified training example based on Theorem 4.1 with the interpolation ratio fixed as $1.0$. One can find that the individual optimal temperatures mostly concentrate between $0.5$ and $1.5$. In Figure 13 right, we solve the optimal interpolation ratio for each incorrectly classified training example based on Theorem 4.2 with the temperature fixed as $1.0$. One can find that the individual optimal interpolation ratio mostly concentrate between $0.5$ and $0.7$.

## E  TOOLKIT

### E.1  ESTIMATION OF THE DATA QUALITY

We use the predicative probabilities of classifiers trained on CIFAR-10 to score its training data. Similar strategy is employed in previous works to select high-quality unlabeled data to improve adversarial robustness (Uesato et al., 2019; Carmon et al., 2019; Gowal et al., 2020). Slightly deviating from these works focusing on out-of-distribution data, we use adversarially trained instead of regularly trained models to measure the quality of in-distribution data, since under standard training almost all training examples will be overfitted and gain overwhelmingly high confidence. Specifically, we adversarially train a pre-activation ResNet-18 with PGD and select the model at the best checkpoint in terms of the robustness. The quality of an example is estimated by the model probability corresponding to the true label without adversarial perturbation and random data augmentation (flipping and clipping). We repeat this process 10 times with random initialization to obtain a relatively accurate estimation.

### E.2  DETERMINE THE OPTIMAL HYPER-PARAMETERS

We employ a model overfitted on the training set to generate approximate traditional adversarial label of the adversarial example in the validation set. Such overfitted model is typically the model at the final checkpoint when conducting regular adversarial training for sufficient epochs. Mathematically, our final method to determine the optimal temperature and interpolation ratio in rectified model probability can be described as

$$T, \lambda = \underset{T, \lambda}{\arg\min} \, \mathbb{E}_{(x_\delta, y_\delta) \sim \mathcal{D}_{\text{val}}} \, \ell\left(\lambda \cdot f(x_\delta; \theta, T) + (1 - \lambda) \cdot f(x_\delta; \theta^s, T), y_\delta\right), \quad (14)$$

where $f(x_\delta; \theta^s, T)$ denotes the predictive probability of a surrogate model scaled by temperature on $x_\delta$.

# F EXPERIMENTAL DETAILS

## F.1 SETTINGS FOR MAIN EXPERIMENT RESULTS

**Dataset.** We include experiment results on CIFAR-10, CIFAR-100, Tiny-ImageNet and SVHN.

**Training setting.** We employ SGD as the optimizer. The batch size is fixed to 128. The momentum and weight decay are set to 0.9 and 0.0005 respectively. Other settings are listed as follows.

- CIFAR-10/CIFAR-100: we conduct the adversarial training for 160 epochs, with the learning rate starting at 0.1 and reduced by a factor of 10 at the 80 and 120 epochs.
- Tiny-ImageNet: we conduct the adversarial training for 80 epochs, with the learning rate starting at 0.1 and reduced by a factor of 10 at the 40 and 60 epochs.
- SVHN: we conduct the adversarial training for 80 epochs, with the learning rate starting at 0.01 (as suggested by (Chen et al., 2021)) and reduced by a factor of 10 at the 40 and 60 epochs.

**Adversary setting.** We conduct adversarial training with $\ell_\infty$ norm-bounded perturbations. We employ adversarial training methods including PGD-AT, TRADES and FGSM. We set the perturbation radius to be $8/255$. For PGD-AT and TRADES, the step size is $2/255$ and the number of attack iterations is 10.

**Robustness evaluation.** We consider the robustness against $\ell_\infty$ norm-bounded adversarial attack with perturbation radius $8/255$. We employ AutoAttack for reliable evaluation. We also include the evaluation results again PGD-1000, Square Attack and RayS.

**Neural architectures.** We include experiments results on pre-activation ResNet-18, WRN-28-5, WRN-34-10 and VGG-19.

**Hardware.** We conduct experiments on NVIDIA Quadro RTX A6000.

## F.2 SETTINGS FOR ANALYZING DOUBLE DESCENT IN ADVERSARIAL TRAINING

**Dataset.** We conduct experiments on the CIFAR-10 dataset, without additional data.

**Training setting.** We conduct the adversarial training for 1000 epochs unless otherwise noted. By default we use the Adam optimizer with the learning rate fixed as 0.0001, since it requires minimal hyper-parameter tuning. For SGD the learning rate starts at 0.1, and will not be changed unless otherwise noted. The batch size will be fixed to 128, and the momentum will be set as 0.9 wherever necessary. No regularization such as weight decay is used. These settings are mostly aligned with the empirical analyse of double descent under standard training (Nakkiran et al., 2020).

**Sample size.** To reduce the computation load demanded by exponential training epochs In individual cases, we reduce the size of the training set by randomly sampled a subset of size 5000 from the original training set without replacement. This will linearly shift the double descent curve but will not significant distort its shape as shown in Appendix B.1. We adopt this setting for extensive experiments such as the analyses of the dependence of double descent on perturbation radius, data quality and the number of attack iterations. Note that in the experiment associated with data quality, we randomly sampled the training subset from those examples with quality lower than a threshold. The sampled subset is restricted to class-balanced.

**Adversary setting.** We conduct adversarial training with $\ell_\infty$ norm-bounded perturbations. We employ standard PGD training with the perturbation radius set to $8/255$ unless otherwise noted. The number of attack iterations is fixed as 10, and the perturbation step size is fixed as $2/255$.

**Robustness evaluation.** We consider the robustness against $\ell_\infty$ norm-bounded adversarial attack with perturbation radius $8/255$. We use PGD attack with 10 attack iterations and step size set to $2/255$.

**Neural architecture.** By default we experiment on Wide ResNet (Zagoruyko & Komodakis, 2016) with depth 28 and widening factor 5 (WRN-28-5) to speed up training.

**Hardware.** We conduct experiments on NVIDIA Quadro RTX A6000.

### F.3 SETTINGS FOR ADVERSARIAL AUGMENTATION

We generate adversarial examples with PGD attack on the model obtained at the best checkpoint through a pratical adversarial training (see Appendix F.1 for details). The number of attack iterations is fixed as 10 and the step size fixed as $2/255$. The adversarial examples of all training examples along with their labels are then grouped into a new training set, where we conduct standard training for 1000 epochs. Other settings are same as those listed in Appendix F.2 except no adversary is employed.

