# OpenReview forum: "Double Descent in Adversarial Training: An Implicit Label Noise Perspective"
_ICLR.cc/2022/Conference — ICLR 2022 Submitted_

### Official Review · Reviewer_JBGf · 2021-10-26

**Correctness:** 4
**Technical Novelty And Significance:** 3
**Empirical Novelty And Significance:** 3
**Recommendation:** 6
**Confidence:** 4

**Main Review:**

The paper is well written and the proposed technique to mitigate the overfitting behavior for adversarial training is clearly motivated in sections 3 and 4. However, I have a few concerns regarding the experimental studies and comparative methods:-

1. Writing: The experimental setup and in general the comparative model is not very well written.


2. Comparative methods:
Chen et al. (2021) [1] proposed to apply a self-training smoothed loss function using both robust-/standard-trained self-teachers (in Eq. 1 of [1]). That required two hyperparameters i.e., $\lambda_1$ and $\lambda_2$. Next, they apply stochastic weight averaging (SWA).

--- However, the authors compared with "PGD training combined with self-distillation (KD-AT) with fixed temperature $T = 2$ and interpolation ratio $\lambda$ = 0.5 as suggested by Chen et al. (2021)." To the best of my understanding, this is a variant of their proposed technique and not the technique proposed in [1]. Can you please comment on this?


3. Comparative results in Table 1:
Chen et al. (2021) already demonstrated to reduce the best vs last robust accuracy to almost close to 0\% (using Auto-attack and CW attack in Table 4 of Chen et al. [1]). However, these results are significantly better than the reported results for comparative models in Table 1.

A few minor comments:-

1. (Section 3.3) "It has been widely observed that adversarial examples transfer between different classifiers even with distinct architectures "

-- I don't think that this comment is true for adversarially trained robust models. It may be better to remove this line.

2. Theorem 4.1 and 4.2: Please mention that the proofs are provided in the appendix.

3. Figure 5(c): why didn't you use the same number of training epochs for TinyImagenet and CIFAR-10/CIFAR-100 datasets?

4. Appendix should have been submitted along with the main paper, after reference.

[1] Tianlong Chen, Zhenyu Zhang, Sijia Liu, Shiyu Chang, and Zhangyang Wang. Robust overfitting may be mitigated by properly learned smoothening. In International Conference on Learning Representations, volume 1, 2021

**Summary Of The Paper:**

This paper demonstrates that "robust overfitting" during adversarial training is an early part of an epoch-wise double descent phenomenon for relatively large models. The authors also demonstrate that the overfitting behavior is observed as we select larger perturbations during training.

The authors explained this "double descent" phenomenon using "implicit label noise" and finally demonstrated that by temperature scaling and knowledge distillation, we can significantly mitigate the overfitting behavior for adversarial training.

================= Post Rebuttal =================
The authors have answered most of my concerns. Hence, I have increased my score to 6.
Please find additional comments to improve the organization of the paper and experimental results in the following.

**Summary Of The Review:**

Pros:- This paper demonstrates that "robust overfitting" during adversarial training is an early part of an epoch-wise double descent phenomenon for relatively large models. Proposed a novel technique motivated by this analysis.

Cons:- Comparative results are not convincing enough.

Overall, I liked the paper. I am giving an initial score of 5 due to my concerns about comparative models in experiments.  If the authors can clarify my concerns, I shall increase my score towards acceptance.

---

> ### Author Response · Authors · 2021-11-23
> **Response to Reviewer JBGf**
>
> We thank the reviewer for the valuable comments. Please check our responses below.
>
>
> ## ____
> **Our method and additional techniques in [1] are orthogonal contributions and can be combined**
>
> We note that motivated by our theoretical analyses, our proposed method amounts to knowledge distillation for adversarial training (KD-AT) with a robustly trained self-teacher, equipped with an algorithm that automatically finds its optimal hyperparameters (i.e. the temperature $T$ and the interpolation ratio $\lambda$), which we now denote as KD-AT-Auto. Stochastic Weight Averaging (SWA) and additional standard teachers (KD-Std) employed in [1] are orthogonal contributions. We note KD-AT-Auto (our method) can certainly be combined with SWA and KD-Std to achieve better performance. To show their synergy, we conducted experiments on the same datasets as [1]. On CIFAR-10, CIFAR-100 and Tiny-ImageNet, KD-AT + KD-Std + SWA [1] can already reduce the overfitting gap (difference between the best and last robust accuracy) to almost 0, while KD-AT-Auto + KD-Std + SWA maintains an overfitting gap close to 0. Interestingly, on the SVHN dataset [2], where KD-AT + KD-Std + SWA still produces a high overfitting gap (see Appendix A1.3 in [1]), KD-AT-Auto + KD-Std + SWA can further push this gap to close to 0 (see below).
>
> | Method | T | $\lambda$ | Robust Acc. (AA) (Best) | Robust Acc. (AA) (Last)| Robust Acc. (AA) (Diff)| Standard Acc. (Best) | Standard Acc. (Last) | Standard Acc. (Diff) |
> | :- | :-: | :-: | :-: | :-: | :-: | :-: | :-: | :-: |
> |PGD-AT | - | - | 47.83 | 39.77 | 8.06 | 90.18 | 91.11 | -0.93 |
> |KD-AT + KD-Std + SWA | 2.0 | 0.5 | 47.88 | 46.46 | 1.42 |  **91.59**  | **91.76**  | -**0.17** |
> |KD-AT-Auto + KD-Std +  SWA | 1.53 | 0.83 | **50.58** | **50.09** | **0.49** | 90.54 | 90.76 | -0.22 |
>
> Furthermore, we note that [1] shows SWA and KD-Std are essential components to mitigate robust overfitting on top of KD-AT, while we show that KD-AT itself can mitigate robust overfitting by proper parameter tuning. We are thus able to separate these components and allow a more flexible selection of hyperparameters in diverse training scenarios without fear of overfitting. In particular, although [1] suggests SWA starting at the first learning rate decay (exactly when the overfitting starts, 80 epoch in our training setting) mitigates robust overfitting, the effectiveness of SWA on mitigating overfitting may strongly depend on the hyper-parameter selection (e.g. when to switch on SWA during training, how to set the decay rate of SWA), which is also mentioned in recent work [3]. We also did some additional experiments on CIFAR-10 following the SWA setting in [3] to demonstrate the wide applicability of our method. As shown by the following table, when changing the hyperparameters of SWA, KD-AT + KD-Std + SWA cannot consistently mitigate robust overfitting, while KD-AT-Auto + KD-Std + SWA can maintain an overfitting gap close to 0 and achieve better robustness as well.
>
> | Method | SWA start at epoch | SWA decay rate | Robust Acc. (AA)  (Best) | Robust Acc. (AA) (Last) | Robust Acc. (AA) (Diff) | Standard Acc. (Best) | Standard Acc. (Last) | Standard Acc. (Diff) |
> | - | :-: | :-: | :-: | :-: | :-: | :-: | :-: | :-: |
> |KD-AT + KD-Std + SWA | 80 | 0.999 | 49.00 | 48.04 | 0.96 | 84.04 |  **86.11** | -2.07 |
> | KD-AT-Auto + KD-Std + SWA | 80 | 0.999 | **49.35** |  **49.25** |  **0.1** |  **85.38**  | 85.91 | -**0.37** |
> ||
> | KD-AT + KD-Std  + SWA | 0 | 0.999 | 49.01 | 48.01 | 1.0 | 83.78 |  **86.20** | -2.42 |
> | KD-AT-Auto + KD-Std + SWA | 0 | 0.999 |  **49.32**  |  **49.25** |  **0.07** |  **84.78**  | 85.48 | -**0.7** |
>
> Therefore, we believe that combining our method and the additional techniques in [1] will be a promising future work. We have added more experiment details in the revised draft (see Section 5 and the associated appendix C.2). We thank the reviewer for pointing this out and we are now able to consolidate our work.
>
> ## ____
> **Our method achieves comparable results combined with additional techniques in [1]**
>
> As mentioned above, our method (KD-AT-Auto) can be further combined with additional techniques in [1] and achieve comparable results as reported in [1]. Here we experiment on CIFAR-10 to compare with our results in the main paper. Note our method can still achieve better results on datasets such as SVHN.
>
> | Method | T | $\lambda$ | Robust Acc. (AA) (Best) | Robust Acc. (AA) (Last) | Robust Acc. (AA) (Diff) | Standard Acc. (AA) (Best) | Standard Acc. (AA) (Last) | Standard Acc. (AA) (Diff) |
> | :-  | :-: |  :-: |  :-: |  :-: |  :-: |  :-: |  :-: |  :-: |
> | KD-AT | 2 |  0.5 | 48.76  | 46.33   | 2.43    | 82.89 | 85.49 | -2.60 |
> | KD-AT-Auto | 1.47 | 0.8 |  49.05   | 48.80 | 0.25  | 84.26 | 84.47 | -**0.21** |
> | KD-AT + KD-Std + SWA [1] | 2.0 | 0.5 |  49.98 | 49.89 | 0.09 | **85.06**   | **85.52** | -0.46 |
> | KD-AT-Auto + KD-Std +SWA | 1.47 | 0.8 | **50.03**  | **50.05** | -**0.02** | 84.69  | 84.91  | -0.22 |

---

> > ### Author Response · Authors · 2021-11-23
> > **Response to Reviewer JBGf (continued)**
> >
> > ## ____
> > **Details of experiment setup**
> >
> > Due to the limited space, we have moved more experiment details to Appendix F.
> >
> > ## ____
> > **The number of training epochs for Tiny-ImageNet:**
> >
> > We follow the experiment settings in [1], where the training epochs for Tiny-ImageNet are cut to half compared to that for CIFAR-10/CIFAR-100.
> >
> > ## ____
> > **Other suggestions on presentation:**
> >
> > We followed your suggestion and have removed this line in Section 3.3.  We have now mentioned that the proofs are provided in the appendix and also submitted the appendix along with the main paper.
> >
> >
> > [1] Robust Overfitting May Be Mitigated By Properly Learned Smoothening. Chen et al., 2021.\
> > [2] Reading digits in natural images with unsupervised feature learning. Netzer et al., 2011.\
> > [3] Fixing Data Augmentation to Improve Adversarial Robustness. Rebuffi et al., 2021.

---

> > > ### Comment · Reviewer_JBGf · 2021-11-26
> > > **Comments after rebuttal**
> > >
> > > I thank the authors for their detailed responses. Overall, I feel that the paper presents an interesting analysis of the double descent phenomena for AT models. Hence, I am increasing my score to 6. However, I still feel that the authors should rearrange the paper to include a little more details to explain their experimental results and analysis in their next updated version. While this was not a major concern, I was still expecting to see such changes in the current updated version of the draft.
> > >
> > > Few recommended changes:-
> > >
> > > 1. The authors should explicitly mention that your method achieves "comparable results" as SWA.
> > >
> > > 2. It is recommended to include the results for SWA without combining them with your model. Right now, we cannot see if combining with your model makes it better or worse. (Also, I should mention that the main contribution of this paper is their analysis for double descent phenomena, and did not focus on outperforming the existing methods.)

---

> > > > ### Author Response · Authors · 2021-11-30
> > > > **Reply to Reviewer JBGf**
> > > >
> > > > We thank the reviewer for the additional comments and careful considerations. We will incorporate the suggestions in the next updated draft.
> > > >
> > > > We note that we have included the results of the method proposed by [1] (i.e. KD-AT +KD-Std + SWA) without combing with our method in the previous responses and Appendix C.2 of the updated draft. Since “KD-AT + KD-Std + SWA” has already reduced the overfitting gap (difference between the last and best performance) on CIFAR-10 to almost 0, combining our method and the method proposed by [1] (i.e. KD-AT-Auto + KD-Std + SWA) achieves comparable results. Nevertheless, on the more challenging dataset SVHN (see Appendix A1.3 in [1]), where “KD-AT + KD-Std + SWA” still produces high robust overfitting, our method can further shrink the overfitting gap to almost 0. Therefore we believe our method is an effective complement to the existing method.
> > > >
> > > > [1] Robust Overfitting May Be Mitigated By Properly Learned Smoothening. Chen et al., 2021.

---

### Official Review · Reviewer_mzDh · 2021-11-01

**Correctness:** 4
**Technical Novelty And Significance:** 2
**Empirical Novelty And Significance:** 2
**Recommendation:** 5
**Confidence:** 4

**Main Review:**

### Strength

* The connection between robust overfitting and model-wise double descent was already observed in the deep double descent paper [1]. But the discussion in the original paper was brief and not intended to make a complete point. In this paper, the authors further connect the epoch-wise double descent to robust overfitting, which is relatively novel.

* The experiments conducted in this paper are extensive. The experiments show that the epoch-wise double descent prevails across different hyperparameters including learning rates, optimizers, architectures and, so on.

* The implicit label noise hypothesis is interesting. The experiments on the perturbation radius, gaussian noise, and data quality help understand the theoretical results. The method motivated to mitigate implicit label noise is sound.

### Weakness

* On the theoretical results: the authors attribute the double descent to the implicit label noise - a distribution shift of the ground truth labels between the adversarial and clean samples. Given that humans cannot distinguish adversarial examples, it is questionable how substantial such a shift can impact. Comparing Figure 1 against [1], the test loss trend of adversarial training with $l_\infty$ norm attack of perturbation $8/255$ is similar to the trend of ERM training with 5-10% label noise, which is quite surprising. Does the implicit noise label really have the same impact with such a level of label noise or is there other fundamental differences that makes the adversarial training more challenging? It would be useful to somehow quantify the implicit label noise and understand how severe it has caused.

* On the experimental results: apart from the vanilla adversarial training, the authors mainly compare their proposed method against the baseline proposed in [2]. However, it seems that the authors didn't compare with a complete version of the method as the stochastic weight averaging seems not to be used in their experiment. Since this is not explicitly mentioned in the paper, please correct me if I miss anything.

* Minor: $J$ is not defined in eq. (3).

[1] Nakkiran, Preetum, et al. "Deep Double Descent: Where Bigger Models and More Data Hurt." International Conference on Learning Representations. 2019.

[2] Chen, Tianlong, et al. "Robust overfitting may be mitigated by properly learned smoothening." International Conference on Learning Representations. 2020.

**Summary Of The Paper:**

This paper studies the double descent phenomenon in the adversarial training, with an aim of explaining the wide observation of robust overfitting. The authors made the connection by showing that with a large enough model, the robust overfitting can be seen as the early stage of an epoch-wise double descent. This observation is confirmed with extensive experiments and analyzed by a proposed metric called the implicit label noise. A method is proposed to mitigate such noise and the consequent overfitting.

**Summary Of The Review:**

This paper offers some novel understanding about robust overfitting, but the findings are not surprising as similar observations have been made in the previous literature on the model-wise double descent. I'm also not fully convinced by the equivalent impact of label noise and implicit label noise on the overfitting. The proposed method seems also not significant and the comparison is not very clearly stated in the paper. For these reasons, I'm currently on the negative side.

---

> ### Author Response · Authors · 2021-11-23
> **Response to Reviewer mzDh**
>
> We thank the reviewer for the valuable comments. Please check our responses below.
>
>
> ## ____
> **Impact of implicit label noise**
>
> First, we would like to note that the adversarial training setting would amplify the impact of the implicit label noise, since it adds perturbations to every training sample. In fact, Theorem 3.1 in our main paper, which lower-bounds the implicit label noise by the distance between the assigned label distribution and the true label distribution, provides a way to intuitively quantify the implicit label noise. Say given an input example, the adversarial perturbation distorts its true label distribution from one-hot to [0.9, 0.1, 0, …], namely the probability mass of the argmax class is only slightly reduced from 1 to 0.9 (this is very likely for those low-quality examples based on human judgment as shown in Figure 2). Then the total variation distance between these distributions is 0.1, which means the implicit label noise is at least 0.1. This is already equivalent to 10% label noise based on the connection between implicit label noise and the (instance-wise) probabilistic definition of label flipping noise (see Remark 3.2 in our paper).
>
> Second, we would like to note that implicit label noise in adversarial training changes in every epoch as the adversarial perturbation in the inner maximization is stochastically generated in each epoch. This stochastic nature further promotes the variance and induces more significant double descent compared to the static label flipping noise typically seen in the literature which is applied to the dataset only once before training. One may note that in the static case of implicit label noise, i.e., the adversarial perturbation is only applied to the dataset once before training, the double descent is indeed less significant (see Figure 3 in our paper).
>
> ## ____
> **Similar observation on model-wise double descent**
>
> It is worth mentioning that although [4] have juxtaposed robust overfitting with double descent in adversarial training, they drew the conclusion that robust overfitting and double descent are separate phenomena (see Section 3.3 in [4]). They observed that adversarially training longer seems to always result in worse performance, while modern generalization theory suggests sufficient model complexity should eventually improve the performance (i.e., the double descent). Works following this understanding thus study robust overfitting independent of double descent.
>
> In this study, we are able to advance this understanding towards the unification of robust overfitting and double descent. Robust overfitting is not an exception to the modern generalization theory, and those existing analyses and techniques in double descent can be readily applied to the study of robust overfitting. Therefore, we believe such a finding goes beyond simply extending model-wise double descent in adversarial training to the epoch-wise regime.

---

> > ### Author Response · Authors · 2021-11-23
> > **Response to Reviewer mzDh (continued)**
> >
> > ## ____
> > **Our method and SWA are orthogonal contributions and can be combined**
> >
> > We note that motivated by our theoretical analyses, our proposed method amounts to knowledge distillation for adversarial training (KD-AT) with a robustly trained self-teacher, equipped with an algorithm that automatically finds its optimal hyperparameters (i.e. the temperature $T$ and the interpolation ratio $\lambda$), which we now denote as KD-AT-Auto. Stochastic Weight Averaging (SWA) and additional standard teachers (KD-Std) employed in [1] are orthogonal contributions. We note KD-AT-Auto (our method) can certainly be combined with SWA and KD-Std to achieve better performance. To show their synergy, we conducted experiments on the same datasets as [1]. On CIFAR-10, CIFAR-100 and Tiny-ImageNet, KD-AT + KD-Std + SWA [1] can already reduce the overfitting gap (difference between the best and last robust accuracy) to almost 0, while KD-AT-Auto + KD-Std + SWA maintains an overfitting gap close to 0. Interestingly, on the SVHN dataset [2], where KD-AT + KD-Std + SWA still produces a high overfitting gap (see Appendix A1.3 in [1]), KD-AT-Auto + KD-Std + SWA can further push this gap to close to 0 (see below).
> >
> > | Method | T | $\lambda$ | Robust Acc. (AA) (Best) | Robust Acc. (AA) (Last)| Robust Acc. (AA) (Diff)| Standard Acc. (Best) | Standard Acc. (Last) | Standard Acc. (Diff) |
> > | :- | :-: | :-: | :-: | :-: | :-: | :-: | :-: | :-: |
> > |PGD-AT | - | - | 47.83 | 39.77 | 8.06 | 90.18 | 91.11 | -0.93 |
> > |KD-AT + KD-Std + SWA | 2.0 | 0.5 | 47.88 | 46.46 | 1.42 |  **91.59**  | **91.76**  | -**0.17** |
> > |KD-AT-Auto + KD-Std +  SWA | 1.53 | 0.83 | **50.58** | **50.09** | **0.49** | 90.54 | 90.76 | -0.22 |
> >
> >
> > Furthermore, we note that [1] shows SWA and KD-Std are essential components to mitigate robust overfitting on top of KD-AT, while we show that KD-AT itself can mitigate robust overfitting by proper parameter tuning. We are thus able to separate these components and allow a more flexible selection of hyperparameters in diverse training scenarios without fear of overfitting. In particular, although [1] suggests SWA starting at the first learning rate decay (exactly when the overfitting starts, 80 epoch in our training setting) mitigates robust overfitting, the effectiveness of SWA on mitigating overfitting may strongly depend on the hyper-parameter selection (e.g. when to switch on SWA during training, how to set the decay rate of SWA), which is also mentioned in recent work [3]. We also did some additional experiments on CIFAR-10 following the SWA setting in [3] to demonstrate the wide applicability of our method. As shown by the following table, when changing the hyperparameters of SWA, KD-AT + KD-Std + SWA cannot consistently mitigate robust overfitting, while KD-AT-Auto + KD-Std + SWA can maintain an overfitting gap close to 0 and achieve better robustness as well.
> >
> > | Method | SWA start at epoch | SWA decay rate | Robust Acc. (AA)  (Best) | Robust Acc. (AA) (Last) | Robust Acc. (AA) (Diff) | Standard Acc. (Best) | Standard Acc. (Last) | Standard Acc. (Diff) |
> > | - | :-: | :-: | :-: | :-: | :-: | :-: | :-: | :-: |
> > |KD-AT + KD-Std + SWA | 80 | 0.999 | 49.00 | 48.04 | 0.96 | 84.04 |  **86.11** | -2.07 |
> > | KD-AT-Auto + KD-Std + SWA | 80 | 0.999 | **49.35** |  **49.25** |  **0.1** |  **85.38**  | 85.91 | -**0.37** |
> > ||
> > | KD-AT + KD-Std  + SWA | 0 | 0.999 | 49.01 | 48.01 | 1.0 | 83.78 |  **86.20** | -2.42 |
> > | KD-AT-Auto + KD-Std + SWA | 0 | 0.999 |  **49.32**  |  **49.25** |  **0.07** |  **84.78**  | 85.48 | -**0.7** |
> >
> > Therefore, we believe that combining our method and the additional techniques in [1] will be a promising future work. We have added more experiment details in the revised draft (see Section 5 and the associated appendix C.2). We thank the reviewer for pointing this out and we are now able to consolidate our work.
> >
> >
> > ## ____
> > **Minor comments:**
> >
> >  In Equation (3) $J$ refers to an event or a subset of the label sample space. We have added the definition in the submission.
> >
> > [1] Robust Overfitting May Be Mitigated By Properly Learned Smoothening. Chen et al., 2021.\
> > [2] Fixing Data Augmentation to Improve Adversarial Robustness. Rebuffi et al., 2021.\
> > [3] Reading digits in natural images with unsupervised feature learning. Netzer et al., 2011.\
> > [4] Overfitting in adversarially robust deep learning. Rice et al., 2020.

---

> > ### Comment · Reviewer_mzDh · 2021-11-27
> > **Thank you for your response. Not all my concerns are fully addressed**
> >
> > I appreciate the clarification on the implicit label noise and the additional experiments with respect to the full version of the baseline [2]. Although the authors re-state their contribution as "KD-AT-Auto itself can mitigate robust overfitting without SWA" and show additional results when SWA is not well-tuned, KD-AT-Auto produces nice results, I'm hardly convinced that this contribution is significant enough, given that "KD-AT + KD-Std + SWA" is better than "KD-AT-Auto" and the reported numbers is not competitive as other methods using PreActResNet-18: https://robustbench.github.io/.
> >
> > > It is worth mentioning that although [4] have juxtaposed robust overfitting with double descent in adversarial training, they drew the conclusion that robust overfitting and double descent are separate phenomena (see Section 3.3 in [4]). They observed that adversarially training longer seems to always result in worse performance, while modern generalization theory suggests sufficient model complexity should eventually improve the performance (i.e., the double descent). Works following this understanding thus study robust overfitting independent of double descent.
> > In this study, we are able to advance this understanding towards the unification of robust overfitting and double descent. Robust overfitting is not an exception to the modern generalization theory, and those existing analyses and techniques in double descent can be readily applied to the study of robust overfitting. Therefore, we believe such a finding goes beyond simply extending model-wise double descent in adversarial training to the epoch-wise regime.
> >
> > [4] proposed to separate model-wise double descent and robust overfitting because of the large gap between the best and final model, which is not the case for large models in a standard training. However, I saw a similar phenomenon in the epoch-wise double descent results - with large number of epochs, the gap still remains. To this end, I'm concerned about the argument on its novelty other than extending the previous finding on model-wise double to the epoch-wise.
> >
> > For these reasons, I would like to keep my current score as it is.

---

> > > ### Author Response · Authors · 2021-11-30
> > > **Reply to Reviewer mzDh**
> > >
> > > We thank the reviewer for the additional comments. Please check our further clarification below.
> > >
> > > ## ____
> > > **KD-AT-Auto shall be viewed as a proof-of-concept method for our major contributions**
> > >
> > > As we summarized at the end of the introduction, **our major contributions are ordered as** (1) showing that robust overfitting shall be viewed as the early part of an epoch-wise double descent, extending the common belief in adversarial training (AT); (2) showing that this double descent in AT may originate from the implicit label noise introduced by improper labeling of adversarial examples in AT practice; and (3) showing that there exists alternative labeling that can be established to provably reduce the implicit label noise and mitigate the robust overfitting.
> > >
> > > KD-AT-Auto shall be viewed as a proof-of-concept solution for “alternative labeling” in the 3rd contribution. It is motivated by the analyses of robust overfitting and aiming at mitigating it, while the performance is not our main aim. We sincerely believe that our major contribution, namely the analyses of the robust overfitting phenomenon, would inspire more future work to further improve the performance.
> > >
> > >
> > > ## ____
> > > **KD-AT-Auto can further mitigate robust overfitting on top of the method proposed by [1]**
> > >
> > > To mitigate robust overfitting, our proposed method (“KD-AT-Auto”) is an effective complement to the existing method, although it is not our major contribution. The benefit is not clearly observed on datasets like CIFAR-10 since the method proposed by [1] (i.e. “KD-AT + KD-Std + SWA”) has already reduced the overfitting gap (difference between the last and best performance) to almost 0. Nevertheless, as we showed in the previous responses and also Appendix C.2 of the updated draft, on more challenging datasets like SVHN (see Appendix A1.3 in [1]), where “KD-AT + KD-Std + SWA” cannot effectively mitigate robust overfitting, **our method can further reduce the overfitting gap to almost 0.**
> > >
> > > In terms of performance, we note that it is not fair to compare our method “KD-AT-Auto” and “KD-AT + KD-Std + SWA” proposed by [1] because “KD-AT-Auto” is only a variant of “KD-AT” with an automatic hyperparameter search. Instead, as we emphasized repeatedly,  “KD-AT-Auto” and the techniques proposed by [1] shall be viewed as orthogonal contributions --- they are not competitors and they can collaborate. As shown in the previous responses and also Appendix C.2 of the updated draft, “KD-AT-Auto + KD-Std +SWA” can achieve comparable performance on datasets such as CIFAR-10 and better performance on datasets such as SVHN, compared to “KD-AT + KD-Std + SWA proposed by [1].
> > >
> > > Finally, we note that almost all those methods on *robustbench* achieving significantly better performance with pre-activation ResNet-18 are using additional data either from extra sources or generated, to the best of our knowledge. Such a technique should be another orthogonal contribution.
> > >
> > > ## ____
> > > **Relation between robust overfitting and double descent in the literature**
> > >
> > > We would like to note that [2] proposes to separate robust overfitting and double descent, not robust overfitting and model-wise double descent. Robust overfitting happens in the epoch-wise regime, thus should be naturally parallel to the model-wise double descent.
> > >
> > > Further, by separating robust overfitting and double descent, [2] means robust overfitting has an effect on the notion of overfitting (or increased model complexity) that is different from double descent. The phenomenon that training longer increases model complexity but results in worse performance contradicts the double descent in modern generalization curves, which states increasing model complexity beyond some interpolation threshold should improve the performance.
> > >
> > > In this paper, the observation that training sufficiently longer should eventually improve the performance is able to resolve this contradiction, which means robust overfitting is not different from double descent. Therefore, we sincerely believe that our findings are novel and will offer new insights to the research community.
> > >
> > > Finally, the gap between the best and last performance (or more generally, the gap between the first minimum and the second minimum considering the epoch-wise double descent when training sufficiently longer) **cannot indicate robust overfitting is an exception to double descent**. Double descent never states the second minimum should be equal to or better than the first minimum. Even in standard training, there are many evidences including the empirical results in [3] showing that the second minimum is not necessarily better when the noise ratio is high (see Figure 10 (a) and (c) in [3]), and also the theoretical results in [4] proving that the second minimum will be worse than the first minimum on a small, noisy dataset (see Appendix 1.3 in [4]).

---

> > > > ### Author Response · Authors · 2021-11-30
> > > > **Reply to Reviewer mzDh (continued)**
> > > >
> > > > [1] Robust Overfitting May Be Mitigated By Properly Learned Smoothening. Chen et al., 2021.\
> > > > [2] Overfitting in adversarially robust deep learning. Rice et al., 2020.\
> > > > [3] Deep Double Descent: Where Bigger Models and More Data Hurt. Nakkiran et al., 2020. \
> > > > [4] Double Trouble in Double Descent: Bias and Variance(s) in the Lazy Regime. d’Ascoli., 2020.

---

### Official Review · Reviewer_YNwA · 2021-11-02

**Correctness:** 3
**Technical Novelty And Significance:** 2
**Empirical Novelty And Significance:** Not applicable
**Recommendation:** 6
**Confidence:** 3

**Main Review:**


Strength:
1. as far as I am concerned, the idea of implicit label noise is interesting and can potentially encourage further investigations that connect the fields of adversarial robustness and learning with noisy data.
2. the author proposed a method using temperature scaling and interpolation to reduce the implicit label noise, hence, mitigating the robust overfitting.
3. Both theoretical investigations and numerical experiments are provided, showing the proposed method can help mitigate robust overfitting on realistic datasets.


Weakness/Concerns:

1. From my viewpoint, claims like "label noise is often essential to explain double descent in standard learning for modern neural architectures" are overstated and misleading.

   In fact, the corresponding citations in the paper (Nakkiran et al., 2020; Yang et al., 2020b) do NOT explain double descent as a result of label noise. Instead, Nakkiran et al., 2020 explain double descent in terms of model misspecification, and Yang et al., 2020b explain double descent in the context of the bias-variance trade-off together with the unimodal-behavior of variance.

   Another claim of such kind is at the beginning of section 3.5, where the authors state, "the effect of label noise on double descent has been rigorously studied based upon both analytical settings ..." After briefly checking the citations, none of them recognize label noise as the cause of double descent (correct me if I'm wrong; I could make a mistake here since I only quickly go through those papers.).

   While adding label noise makes the double descent phenomenon more evident, the key concept is still the interplay between the complexity of the model and data. The author should rephrase relevant statements to make them more rigorous.

2. The way of presenting definition 3.2 could potentially cause confusion, and I was confused when I read this part first time. The terminology true label and assigned label reminded me $\arg\max_jP(Y_\delta^*=j|x_\delta)$ and $\arg\max_jP(Y^*=j|x)$, which, according to section 3.2, should be the same, thus, making me wonder why there is a need to consider $P(\tilde{y}_\delta \ne y^*_\delta|x_\delta)$. On the other hand, if I understand correctly, the phrase "its assigned label is different from its true label" in definition 3.2 refers to a "potential difference" that resulted from the mismatch between the true label and assigned label distributions.







**Summary Of The Paper:**

This paper proposes identifying the robust overfitting as the early part of epoch-wise double-descent, which is caused by implicit label noise derived from the mismatch between the true label and assigned label distributions on the adversarial examples. The authors further provide a method that combines temperature scaling and interpolation to mitigate robust overfitting. Both theoretical and experiments are included to show the effectiveness of the proposed method on realistic datasets.


**Summary Of The Review:**

This paper studied the cause of robust overfitting and proposed a method to mitigate it. Both theoretical investigation and experiments on realistic datasets are included for demonstrating the effectiveness of the method. The paper is overall well written. However, the presentation of key concepts needs to be modified, and overstated claims should be rephrased. The key idea of implicit label noise is novel and insightful and could inspire works that connect the fields of adversarial robustness and learning with noisy data.

---

> ### Author Response · Authors · 2021-11-23
> **Response to Reviewer YNwA**
>
> We thank the reviewer for the valuable comments. Please check our responses below.
>
> ## ____
> **The relationship between Implicit label noise and double descent in adversarial training**
>
> We agree with the reviewer that the statement of the relationship between implicit label noise and double descent in adversarial training is indeed not explicit enough. Based on the popular bias-variance understanding of double descent [1, 2], a rigorous statement should be the label noise increases the variance of the model and thus makes the double descent evident. We thank the reviewer for this valuable suggestion and have revised relevant statements in the paper. We list the corresponding revisions as follows.
>
> * Beginning of Section 3.2:
>     * "label noise is often essential to explain double descent in standard learning for modern neural architectures”
>     * →  “In standard learning, it is often necessary to manually inject label noise to make the double descent evident for modern neural architectures. ” (Rephrased from [1])
> * Section 3.5:
>     * “It can be inferred that implicit label noise will cause double descent in adversarial training.”
>     * →  “It can be inferred that implicit label noise will increase the variance and make an evident double descent in adversarial training.”
>
> ## ____
> **Presentation of Definition 3.2**
>
> In Definition 3.2, we define the implicit label noise as “the probability that an input’s assigned label is different from its true label”. Here the label noise should be interpreted probabilistically due to the subtle distribution mismatch. For a specific input $x$, “its assigned label is likely to be different from its argmax true label” should be different from the fact that “its assigned label in the dataset is the same as its argmax true label”, since the former “assigned label” refers to a probability distribution while the latter “assigned label” refers to an outcome (just one sample). To understand this nuance clearly, one can motivate from the frequentist’s perspective and consider a hypothetical situation where there are 100 identical copies of input $x$ in a dataset and all of them have assigned the label “class 0” in a binary classification problem. Now say initially the true label distribution of $x$ is [1, 0], and after adversarial perturbation, it becomes [0.9, 0.1]. Then one should expect 90 copies are now assigned label “class 0” and 10 copies are assigned label “class 1”. However, in adversarial training, we just keep all the original assigned labels and thus create 10% label noise. Here, the assigned label of every copy is the same as the argmax true label, but the label noise still exists in the population. In a realistic dataset, one can consider those inputs with the same or similar true label distribution as a population instead of exact copies.
>
> We note that a similar illustration example has been presented at the end of Section 3.3 in our paper. We have added more notes in Definition 3.2 to clarify the definition of implicit label noise.
>
> [1] Rethinking Bias-Variance Trade-off for Generalization of Neural Networks. Yang et al., 2020.\
> [2] Double Trouble in Double Descent: Bias and Variance(s) in the Lazy Regime. d’Ascoli., 2020.

---

### Official Review · Reviewer_nwPt · 2021-11-02

**Correctness:** 3
**Technical Novelty And Significance:** 4
**Empirical Novelty And Significance:** 4
**Recommendation:** 6
**Confidence:** 5

**Main Review:**

Strengths :
1. This work provides a novel and interesting analysis for explaining the robust overfitting problem of adversarial training. In this work, they make a connection between the double descent in learning with noisy labels and the robust overfitting in adversarial training by introducing “implicit noisy labels”. I believe the novel analysis will provide a new perspective for the community to explain some other phenomenons.
2. This paper is well-written and easy-to-follow. The authors give a clear analysis from the perspective of implicit noisy labels and further propose a simple method based on the analysis.
3. The evaluation of the proposed method is conducted with Autoattack, which makes this work more convincing.

Weaknesses :
Although the analysis sounds pretty interesting, the support for the explanation is somewhat weak. In particular, the authors claim that double descent in adversarial training is caused by implicit noisy labels and empirically verify it by showing that training with static adversarial examples would also encounter the double descent phenomenon. In my opinion, the causal relationship between double descent and implicit noisy labels is not clearly supported.

**Summary Of The Paper:**

In this work, the authors aim to show that double descent in adversarial training might be caused by implicit label noise, that is, the distribution mismatch between the true label distribution and the assigned label distribution of the adversarial examples. They empirically support this claim by showing that training with static adversarial examples would also encounter the double descent phenomenon. To further solve this problem, they propose to apply temperature scaling and interpolation to create a soft label for each adversarial sample in adversarial training. Experiments on CIFAR10/100 and Tiny-imagenet are conducted to validate the efficacy of the proposed method.

**Summary Of The Review:**

Overall, the analysis in this work is novel and interesting, although there exists a weakness in supporting the claim. In this concern, I recommend a score marginally above the acceptance threshold

---

> ### Author Response · Authors · 2021-11-23
> **Response to Reviewer nwPt**
>
> We thank the reviewer for the valuable comments. Please check our responses below.
>
> ## ____
> **The relationship between implicit label noise and double descent in adversarial training**
>
> We note that in the original submission the statement of the relationship between implicit label noise and double descent in adversarial training is not explicit enough. Based on the popular bias-variance understanding of double descent [1, 2], a rigorous statement should be “implicit label noise makes double descent in adversarial training more evident”, since (1) implicit label noise is a specific type of label noise (2) label noise increases the variance of the model and thus makes the double descent evident. Here (1) has been rigorously shown in our paper and (2) has been rigorously proven in [1, 2].  We believe our revised statement of the relationship should be well supported.
> We have carefully revised the relevant statements in the paper and now it should have a more explicit presentation. We list the corresponding revisions as follows.
>
> * Beginning of Section 3.2:
>     * "label noise is often essential to explain double descent in standard learning for modern neural architectures”
>     *    →  “In standard learning, it is often necessary to manually inject label noise to make the double descent evident for modern neural architectures. ” (Rephrased from [1])
>
> * Section 3.5:
>     * “It can be inferred that implicit label noise will cause double descent in adversarial training.”
>     * →  “It can be inferred that implicit label noise will increase the variance and make an evident double descent in adversarial training.”
>
> Please note that in this paper, we aim to better understand the overfitting in adversarial training (i.e., robust overfitting) from a label noise perspective. Our proposed implicit label noise is able to bridge the robust overfitting and the double descent in modern generalization theory. We believe how label noise causes double descent in generalization theory is an orthogonal problem that has been extensively studied in the literature.
>
> [1] Rethinking Bias-Variance Trade-off for Generalization of Neural Networks. Yang et al., 2020.\
> [2] Double Trouble in Double Descent: Bias and Variance(s) in the Lazy Regime. d’Ascoli., 2020.

---

> > ### Comment · Reviewer_nwPt · 2021-12-01
> > **How do the empirical results in Figure 3 support the claims?**
> >
> > I appreciate the clarification about the relationship between implicit label noise and double descent in adversarial training. However, it is still not clear how the results in Figure 3 support the claims. The authors may need to give a detailed explanation for this concern.

---

### Decision · Program_Chairs · 2022-01-20

**Decision:**

Reject

**Comment:**

The paper suggests that robust overfitting could be viewed as the early-part of a double descent phenomenon for adversarial training. The authors identify implicit label noise, i.e. the label distribution mismatch between the true example and the generated adversarial example as a possible explanation for this phenomenon in adversarial training. This claim is empirically supported by experiments using static adversarial examples. The authors propose a method using temperature scaling and interpolation to mitigate the effects caused by implicit label noise for robust overfitting. This method is evaluated on CIFAR 10/100 and tiny-Imagenet. Concerns have been raised in the reviews about sufficient justification for the claim that implicit label noise leads to adversarial overfitting. The rebuttal answers this question to some extent. Concerns have also be raised about the writing and whether sufficient details of the experimental setup are present in the main paper. While I acknowledge the difficulty of fitting all details within page limits, I would think that these details are crucial given that primary support for the claims made are from empirical observations.